# A Smart Many-Core Implementation of a Motion Planning Framework along a Reference Path for Autonomous Cars

**Gianpiero Cabodi** [1] , **Paolo Camurati** [1] , **Alessandro Garbo** [1,†] , **Michele Giorelli** [2] , **Stefano Quer** [1,*,†] and **Francesco Savarese** [1,†]

[1] Dipartimento di Automatica e Informatica, Politecnico di Torino, I-10129 Turin, Italy; gianpiero.cabodi@polito.it (G.C.); paolo.camurati@polito.it (P.C.); alessandro.garbo@polito.it (A.G.); francesco.savarese@polito.it (F.S.)
[2] Automated Driving Technologies, Technology Innovation, Magneti Marelli, Venaria Reale, I-10078 Turin, Italy; michele.giorelli@magnetimarelli.com
* Correspondence: stefano.quer@polito.it
† These authors contributed equally to this work.

**Abstract:** Research on autonomous cars, early intensified in the 1990s, is becoming one of the main research paths in automotive industry. Recent works use Rapidly-exploring Random Trees to explore the state space along a given reference path, and to compute the minimum time collision-free path in real time. Those methods do not require good approximations of the reference path, they are able to cope with discontinuous routes, they are capable of navigating in realistic traffic scenarios, and they derive their power from an extensive computational effort directed to improve the quality of the trajectory from step to step. In this paper, we focus on re-engineering an existing state-of-the-art sequential algorithm to obtain a CUDA-based GPGPU (General Purpose Graphics Processing Units) implementation. To do that, we show how to partition the original algorithm among several working threads running on the GPU, how to propagate information among threads, and how to synchronize those threads. We also give detailed evidence on how to organize memory transfers between the CPU and the GPU (and among different CUDA kernels) such that planning times are optimized and the available memory is not exceeded while storing massive amounts of fuse data. To sum up, in our application the GPU is used for all main operations, the entire application is developed in the CUDA language, and specific attention is paid to concurrency, synchronization, and data communication. We run experiments on several real scenarios, comparing the GPU implementation with the CPU one in terms of the quality of the generated paths and in terms of computation (wall-clock) times. The results of our experiments show that embedded GPUs can be used as an enabler for real-time applications of computationally expensive planning approaches.

**Keywords:** software engineering; many-core architectures; concurrent programming; path planning; autonomous driving

## 1. Introduction

Autonomous driving systems are becoming more real in our daily life, and new techniques and new improvements are proposed by researchers and companies at a high rate. An autonomous car has to fulfill many tasks, working in a highly dynamic environment, respecting hard real-time constraints, and minimizing error probability. In particular, the local planner module is highly "sensible" to driver choices, environment changes, and time constraints. This module is the one responsible for

generating several different trajectories and selecting the best one to follow. The selected path has to be collision-free, suitable for the vehicle, and the most cost-effective with respect to a given cost function.

When path finding and graph traversal are concerned, computer scientists often refer to A\*, due to its widespread use, its performance, and its accuracy. Hart et al. [1] proposed it in 1968 to build a mobile robot that could plan its own actions. It can be seen as an extension of Dijkstra's algorithm. The main difference from A\* to other greedy best-first search strategies is that it takes the cost (or distance) already traveled into account together with the estimated distance from the current position to the goal. Modern techniques approach trajectory generation by either sampling the state space or by sampling the control space. Both strategies have advantages and disadvantages. Techniques sampling the state space [2] design the planning looking first at the environment and choosing collision-free points in the space. Only in a second stage they compute a feasible trajectory. In this way, trajectories are for sure collision-free, but it is not guaranteed that they are also physically feasible for the vehicle. Techniques sampling the control space [3,4] select arrival points in the space looking at suitable vehicle controls. As a consequence, the trajectories are guaranteed to be feasible by construction, but the sampling process can lead to collisions as it does not take into account the space configuration. Both research paths have been successfully applied, proving their effectiveness, and several studies have been conducted to improve the quality of the generated trajectories.

*Sample-based* planning techniques [5–7] sample the configuration space into a set of finite motion goals. *Randomized sample-based* algorithms generate those goals randomly in close proximity of a reference path (or a trajectory) such as a road or a lane infrastructure. *Rapidly-exploring Random Trees* (RRT) [8–10] have recently been adopted with complex environments due to their ability to search high-dimensional input spaces, to navigate among static and dynamic obstacles, and to consider vehicle dynamics and terrain shape in their solution. Given a vehicle, these algorithms generate a set of paths to explore the state space along a given reference path. This set is organized as a tree, where the root node is placed in the initial vehicle position, vehicle trajectories are represented by tree paths from root to leaves, and each new tree level explores the space for a specific distance or time. Among all generated paths from root to leaves, these techniques eventually select the best path, using a proper cost function. The cost function usually evaluates the distance of each leaf from the desired target trajectory, the path geometry (feasibility and convenience of the trajectory), and its safety (distance from all fixed and moving objects along the path). Among the advantages, these methods do not require good approximations of the reference path, they do not rely on accurate vehicle models, and they derive their power from an extensive computational effort directed to improve the quality of the trajectory from step to step.

In this work, we concentrate on how to exploit the power of recently born massive parallel architectures to improve the efficiency and the quality of existing algorithms, rather than developing new ones. Indeed, we focus our attention on how to re-engineer the randomized sample-based algorithm presented by Schwesinger et al. [10], on a CUDA (a parallel computing platform and application programming interface model created by Nvidia, acronym for Compute Unified Device Architecture) many-core embedded GPGPU-based architecture.

First of all, while transforming the sequential algorithm into a concurrent one, we trimmed the method to its best, by optimizing all parameters and each basic step. Then, we re-implemented the original algorithm using different CUDA kernels, each one running several working threads on the GPU. We defined how to propagate information among threads, how to synchronize those threads, and how to organize memory transfers between the CPU and the GPU (and among the different CUDA kernels) to reduce planning times and to minimize the memory usage. We finally obtained an application completely implemented in CUDA, and running all path-planning activities on an embedded GPU.

Overall, our main motivations are the following ones. First of all, planning tasks need efficiency and scalability which modern embedded CPUs may fail to guarantee. CPUs may become a bottleneck on automotive applications given their overall working load and the generated and dissipated power.

We are also motivated by the purpose to adopt the CUDA technology as a de facto standard in many applications that enable efficient many-core implementations. Thirdly, though many-core applications are becoming a standard, efficient design and implementation techniques have not yet spread enough among researchers in the field. Overall, we present the approach we adopt to port a CPU-based algorithm to a many-core GPU-based platform.

In the experimental result section, we compare our sequential CPU-based implementation of the method presented by Schwesinger et al. [10] with a many-core concurrent GPU-based one. In this respect, explicitly comparing our GPU implementation with any other one would be meaningless. This sort of comparison would grade the quality of the original algorithm against other similar or dissimilar methodologies, while our target is to identify the potential benefit of a GPU implementation against the corresponding CPU one. We define an evaluation framework and quantitative metrics, going beyond the mere wall-clock time required to run the algorithm, to evaluate our implementation in several critical scenarios. We present the impact that our set-up may have on real world scenarios. We also analyze several critical aspects of the original algorithm, as well as of our parallel CUDA implementation.

Power consumption and energy efficiency are undoubtedly key aspects to consider when resorting to GPUs, especially when they may replace CPUs. The issue has been addressed in numerous papers, covering aspects of power measurements and/or estimation. A recent survey on models and tools for measurement and estimation of GPU power consumption appears in [11]. However, the debate among researchers is still open as whether GPUs are more power- and energy requiring than CPUs or not, and under which conditions. This paper does not address the issue, as we are not considering the final hardware platforms, rather we work on an industrial prototype including both a multi-core CPU and a GPU. Although required for the final hardware and software architecture evaluation, CPU versus GPU comparison on power efficiency is thus beyond the scope of this work.

As a final remark, notice that the project has been developed under an industrial non-disclosure agreement between Politecnico di Torino and Magneti Marelli. For that reason, the software and the experiments cannot be made publicly available.

### 1.1. Contributions

Graphical Processing Units have been spreading in many scientific domains, and they have also been used in some recent works on automotive motion planning (see Section 3.2 for further details on this issue). However, those works usually present a new planning algorithm, or some new feature of an existing planning strategy. In all cases, GPUs are used to run only specific and particularly expensive phases of the planner. Moreover, those strategies do not specifically concentrate on the parallel implementation and often do not apply any specific memory management optimization. We concentrate our analysis on how a fully GPU-based implementation can be obtained starting from an existing state-of-the-art sequential application. Our parallel algorithm for path planning is entirely implemented on a CUDA GPGPU environment. As far as we know, we are the first to describe into details such a process. Furthermore, we present detailed results on how our GPU-based approach compares with our standard CPU-based one, in terms of both path accuracy and time efficiency.

To sum up, our major contributions are the following:

- We analyze the original algorithm in different scenarios to understand the quality of the original path planner and trim it to its best.
- We implement several concurrent versions of the original algorithm, to analyze transfer, memory, and computational time issues.
- We compare CPU and GPU results, performing tests on several real maneuvers. Results are analyzed defining, and computing, specific ad-hoc metrics and evaluation parameters.
- We show that paths obtained using the GPU are completely superimposable to the ones collected with the CPU. At the same time, response times are drastically reduced, giving the system

enough time to span the space deeper and more accurately or to improve the system's behavior in critical conditions.

## 2. Background

### 2.1. Path Planning

*Planning* has different meanings according to the domain of application [12]. In this paper, we refer to the meaning used in robotics. One major goal in robotics is the conversion of high-level tasks, frequently expressed in natural language, into low-level descriptions of how to move. The term *Path Planning* includes all the algorithms capable of performing this task. It also entails interaction with kinematic constraints, dynamics, obstacle avoidance, and so on.

As shown in Figure 1 path planning is generally structured in three hierarchical modules: a *Global Planner*, a *Decision Maker*, and a *Local Planner*.

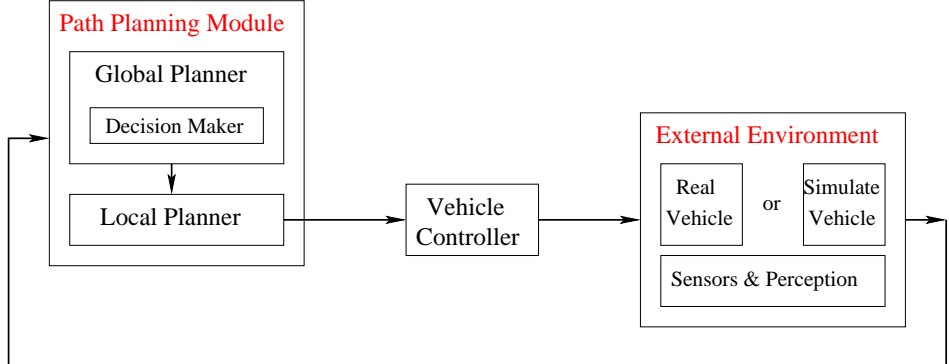

**Figure 1.** Path Planner Structure (global planner, decision maker, and local planner) and its relationship with the vehicle and the external environment.

The global planner is in charge of generating a long-term path based on user requests. The decision maker creates short-term paths also called maneuvers, such as lane change, vehicle overtaking and vehicle following. The local planner creates the trajectory, i.e., the set of points the vehicle should follow to implement the maneuver.

Trajectories are fed to a vehicle controller, represented in Figure 1 with the box just outside the planner. The controller transforms the trajectory into suitable commands to turn and to accelerate/decelerate the vehicle. During the development phase, real vehicles can be replaced by a simulator. The planner receives pre-processed data from the vehicle (or the simulator), such as maps of the environment including obstacles and constraints, speed, position and so on.

### 2.2. GPGPU and Parallel Programming Basic Notions

General purpose graphical processor units (GPGPU) are becoming more and more invasive in every-day life [13–18].

They are especially well-suited to address problems that can be expressed as data-parallel computations where many concurrent threads run in parallel. As all threads execute the same code, CUDA programming is an instance of the well-known SPMD (single-program, multiple-data) parallel programming style, a popular programming style for massively parallel computing systems derived from SIMT (single-instruction, multiple-threads) architectures.

CUDA C extends C by allowing the programmer to define C functions, called kernels, that, when called, are executed $N$ times in parallel by $N$ different CUDA threads, as opposed to only once like regular C functions. A kernel is defined using a special CUDA declaration, and it is executed using the $<<< \ldots >>>$ construct. For the sake of simplicity, we will use $< \ldots >$ in our pseudo-code to represent the same operation. Each thread that executes the kernel is given a unique thread ID

that is accessible within the kernel through a 3-component built-in variable. This provides a natural way to invoke computation across the elements in a domain such as a vector, matrix, or volume. However, a kernel can be executed by multiple equally-shaped thread *blocks*. As a consequence, the total number of threads running is equal to the number of blocks multiplied by the number of threads per block. Following CUDA, we will specify the number of blocks and the number of threads per block as $< numberOfBlock, numberOfThreadsPerBlock >$. Blocks are organized into a one-dimensional, two-dimensional, or three-dimensional grid of thread blocks.

As a final consideration, notice that by using GPU and CUDA we can distinguish among several types of memory spaces (from the one with the smallest latency time on):

- Constant memory (for read-only constant data), registers, and private local memory space. This is local to each thread.
- Shared memory spaces, accessible only by threads in the same block.
- Global memory, based on texture cache visible to all grid threads.
- Global memory, accessible in read-and-write mode by all threads.

## 3. Related Works

### 3.1. Path Planning Methodologies

Referring to Figure 1, local planners represent the most sensible section of the entire path planning module, as local trajectories are subject to stringent mathematical constraints that prevent purely mathematical solutions. In order to solve this issue, several algorithms and frameworks have been proposed, such as *sample-based* (*randomized* and *deterministic*) techniques, *control* strategies, and *purely geometrical* planning methods.

Instead of exploiting the configuration space in a continuous way, *sample-based* planning techniques [5–7,10] sample the configuration space into a set of finite motion goals. Although the sampling phase could affect the optimality of the final trajectory, it allows significant speed up and more realistically meet real time constraints. Furthermore, sample-based techniques do not need sophisticated mathematical approaches.

Within the framework of sample-based planning, *randomized* algorithms are mostly based on the so-called Rapidly-exploring Random Tree (RRT) [5,6]. In this case the driving idea is to iteratively expand a random tree by applying control inputs that drive the system toward randomly-selected points. Originally, if online operations were requested, these methods were restricted to approximately four-dimensional state spaces, thereby limiting the fidelity of the models they could reproduce. They thus mainly found application in situations where no external structure could be extracted for guidance, such as off-road or large-scale parking lots. If a reference path (or a trajectory) is available for guidance, such as road or lane information, a common approach is to align the end-points of local trajectory samples with the reference path (see for example Werling et al. [19]). This technique simultaneously reduces search complexity, and overcomes the danger of entering unsafe states. Most of these approaches are based on geometric primitives instead of actual vehicle models, separating the computation of velocity profiles from the geometric construction of the path [20–22]. This separation may result in conflicts, especially at low speeds. Consequently, trajectories need to be validated in a post-processing step, which may in turn lead to the pruning of a considerable amount of candidate motion [19].

*Deterministic sample-based algorithms* [10,23,24] explore the configuration space without applying any probabilistic function. For example, in graph-based approaches [23] both speed and space are completely discretized into a finite set of samples with a certain resolution.

*Control strategies* [25,26] represent a natural formalism for representing path problems. In these methods the planning process is expressed through differential equations, which take into consideration the initial and the terminal states, the mathematical model of the vehicle, the cost function that should be minimized, and practical constraints that have to be taken into consideration

(such as limitation on the actuators). This continuous-time optimization problem can be transformed into a nonlinear programming task assuming a parametric solution. Unfortunately, this high-level notation masks severe difficulties. First of all, generally there are no analytic solutions to compute first derivatives with respect to the parameters, and implementations must rely on numerical methods. Moreover, numerical strategies may be trapped in local minima. Furthermore, they are really time-consuming, and thus unsuitable for real-time applications.

Purely *geometric planning techniques* [27] represent the oldest planning approaches, developed especially for mobile robots. To produce smooth trajectories, with respect to the comfort of human body, these techniques transform the planning problem into an interpolation task. The main advantages of these planning techniques are their low computational cost and the continuity ensured for the control inputs. However, these approaches seem to be too simple and inappropriate for real applications, as trajectories turn to be neither feasible nor optimal. Furthermore, collision avoidance methods are often not integrated with these algorithms since no different candidate trajectories are offered.

## 3.2. GPU-Based Path Planning Strategies

GPU analysis has also been the subject of some recent works on automotive motion planning.

Pan et al. [28] introduce a motion planning randomized algorithm that exploits the computational capabilities of many-core GPUs. This approach uses threads and data parallelism to achieve high performance for all components of sample-based algorithms, including random sampling, nearest neighbor computation, local planning, collision queries and graph search. The authors demonstrate the efficiency of their algorithm by applying it to several 6 degrees-of-freedom planning benchmarks in 3D environments.

Kider et al. [29] implement a randomized variant of the $A^*$ algorithm. The core of the search is transformed into a CUDA kernel. They test their parallel algorithm using a 6 degrees-of-freedom planar robotic arm showing that their GPU-based approach offers significant improvements in term of solution cost and chances of finding feasible solutions.

McNaughton et al. [30] suggest a search space representation that allows the search algorithm to systematically and efficiently explore both spatial and temporal dimensions in real time. Their main contribution is how to solve a high dimensional state space optimization problem using an exhaustive search algorithm. As this cannot be done on a CPU multi-threading environment without specific optimizations, the authors moved the complete state transition and lattice generation onto the GPU device. The authors show that their algorithm could readily be accelerated on a GPU, and demonstrate it on an autonomous passenger vehicle.

As real-time constraints for path planning are often quite tight, Heinrich et al. [31] present a sampling-based planning method considering motion uncertainty to generate more human-like driving paths. Given information in the form of a small set of rules and driving heuristics, the planning system optimizes trajectories in a seven-dimensional state space. Results show that a mobile GPU can be used as an enabler for real-time applications of computationally expensive planning approaches.

Notice, that as already described in Section 1.1, all previous works concentrate on new algorithmic features, and essentially use a GPU to enable real-time computations. In fact, only specific expensive sections of the planner are usually run on the GPU, and many methods do not apply any specific memory management optimization.

## 4. Base Line Algorithm

Starting with a list of terms used in this paper, in order to make it self-contained, this section summarizes the algorithm presented by Schwesinger et al. [10].

## 4.1. Terminology

Referring to Figure 2:

- The *Local Path* is a finite set of points that the vehicle should ideally follow.
- The *Local Vehicle State* (LVS) is the description of the current vehicle state. It is a tuple $(x, y, \theta, v)$ where $x$ and $y$ are the vehicle coordinates in a 2D space, $\theta$ is the vehicle orientation, and $v$ is the vehicle speed.
- The *Lookahead Time* ($T_{lookahead}$) is a parameter of the algorithm that drives the identification of terminal states while generating trajectories, i.e., the time difference between the terminal states and the current time ($T_0$).
- The *Simulation Time* ($T_{sim}$) is the parameter in charge of regulating the density of the points computed by the algorithm that form the trajectory in a discrete way.
- The *Planning Time* ($T_{cycle}$) is the time needed by the local planner to compute the trajectory. Its value represents the temporization for trajectories generation and it is strictly bounded by real time environment constraints. We usually work with $T_{cycle} = 20$ ms.
- The *Occupancy Grid Map* is a discrete representation for free and occupied portions of the space. Grid maps are the standard model for environment representation in mobile robotics [32–35], and are used for collision avoidance and for trajectory cost evaluation. In Figure 2 black tones are used to represent obstacles. They become first gray and then white as distance from those objects increases.

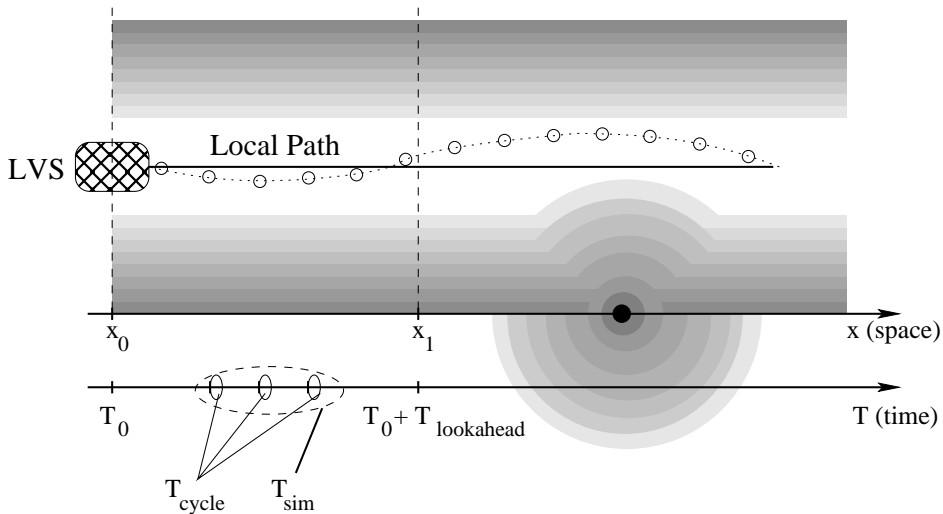

**Figure 2.** A graphical representation for our terms on a standard background occupancy grid map. Black areas represent obstacles which must be avoided at all costs. Gray tones become darker closer to black areas to represent an increasing level of danger. The vehicle position is represented at time $T_0$ in horizontal position $x_0$. The current vehicle direction is represented by the horizontal black line, whereas the local path (indicated by the dotted line) is computed to maintain the vehicle at the center of the white area, thus minimizing the risk of collision. Dots on the local path represent position samples. The algorithm will target the terminal states computed at time $T_0 + T_{lookahead}$ and horizontal position $x_1$.

Please notice that in our environment, occupancy grids are created by a data fusion system resorting to data coming from vision, GPS, radar and LiDAR sensors. We will not discuss how local paths and occupancy grids are generated as these topics are outside the scope of the local path planner and of our presentation as well.

*4.2. The Algorithm*

The local planner generates an optimal trajectory as a result of a planning cycle. Within each run of the planning cycle, the planner generates a set of trajectories, organized as a tree as represented in Figure 3. The tree is built level by level. At each level, the algorithm tries to explore (reach) a larger set of objectives starting from the current set of possible vehicle positions. While the initial position (for

the first tree level) coincides with the initial vehicle coordinates, each new objective is computed using a predefined (node or) path splitting policy. Objectives are found guessing the desired vehicle position after $T_{lookahead}$ time units, considering differing vehicle lateral offsets (A lateral offset is a position displacement used to sample the space around the vehicle position prediction during tree building) and vehicle longitudinal speeds. Each tree level spans the space for $(T_{lookahead}/H)$ time units, where $H$ is the tree height. At tree level 0 the current position is unique and the number of objectives is equal to the number of root children $D$. At level 1, there are $D$ current positions and $D^2$ targets, etc. The entire process is repeated $H$ times, generating a tree with $H$ levels. Leaves are then at a $T_{lookahead}$ time unit distance from the initial position. The algorithm finally applies to all trajectories a cost function, to select and return the best one according to given criteria. As it is not guaranteed that objectives are indeed reached, the outcome is the closest feasible tree node, and the corresponding edge, leading from the tree root to the best first level node.

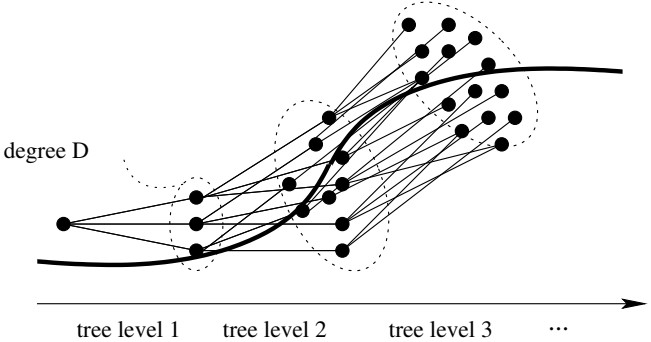

**Figure 3.** A random tree exploring the area around a given trajectory. In the representation, the degree of the tree is equal to $D = 3$, so is its height $H$.

As shown in Algorithm 1, tree nodes represent pairs (LVS, timestamp) while tree edges are sets of points connecting parent and child LVS pairs. Nodes and edges are collected in $N$ and $E$, respectively. Along the reference path, a set of terminal states, $M$, is built from a discrete set of lateral offsets $O$, and a set of longitudinal vehicle velocities $V$. The tree degree $D$ is equal to $D = |O| \times |V|$. Variables $t_0$, $\widehat{n}$, $\widehat{e}$, and $u_0$ represent the initial time-stamp, the computed node and edge, and current vehicle commands, respectively. Parameter $f$ represent the vehicle model. Parameter $g$ is a user-supplied controller that controls the simulated system model parameter $f$ (see Schwesinger et al. [10] for further details).

In Algorithm 1, lines 1–2 initialize the data structures. To take planning time $T_{cycle}$ into account, function SIMULATE, at line 3, expands the root vertex $n$ generating $\widehat{n}$, i.e., the LVS at time-stamp $(T_0 + T_{cycle})$, and $\widehat{e}$. Queue $Q$, initialized to $\widehat{n}$ at line 5, supports breadth-first tree building. Lines 6–16 are the core of the algorithm. Lines 6, 7 and 9 control the expansion of the tree. The outer loop expands the tree to a desired height $H$. The intermediate one iterates on the children of each node. The inner loop expands each node to its degree $D$. Within the inner iteration function DRAWSAMPLE generates $D$ pairs of reference points and speeds $(d_{ref}, v_{ref})$ taking into account the $T_{lookahead}$ parameter. Function EXPAND generates the node closest to each $(d_{ref}, v_{ref})$ pair, and the corresponding edge following the controlled kinematic vehicle model. In lines 17 function COMPUTECOST selects the less expensive trajectory. The minimum cost node $\widehat{n}_{opt}$ (line 18) is identified and the first edge $\widehat{e}_{opt}$ (line 19) leading to it is returned (line 20).

In our version of the algorithm node cost computation (function COMPUTECOST) is performed during the expansion process. Costs propagate from node to node until leaves are reached. Occupancy grid maps serve both the purpose of optimizing cost and of avoiding obstacles.

---

**Algorithm 1** Top-level Local Planner Algorithm.

PLANNING CYCLE

1: $N = \emptyset, E = \emptyset$
2: $n = (LVS, T_0), N = N \cup \{n\}$
3: $(\widehat{n}, \widehat{e}) = $ SIMULATE $(n, f, g, T_{cycle}, u_0)$
4: $N = N \cup \{\widehat{n}\}, E = E \cup \{\widehat{e}\}$
5: Q.insert$(\widehat{n})$
6: **for** $d = 0$ to $H - 1$ **do**
7: 　　**for all** nodes at tree depth $d$ **do**
8: 　　　　$n = Q.extract()$
9: 　　　　**for** $j = 1$ to $D$ **do**
10: 　　　　　$(d_{ref}, v_{ref}) = $ DRAWSAMPLE $(M)$
11: 　　　　　$(\widehat{n}, \widehat{e}) = $ EXPAND $(n, f, g, d_{ref}, v_{ref}, T_{sim})$
12: 　　　　　$N = N \cup \{\widehat{n}\}, E = E \cup \{\widehat{e}\}$
13: 　　　　　$Q.insert(\widehat{n})$
14: 　　　　**end for**
15: 　　**end for**
16: **end for**
17: COMPUTECOST$(E, N)$
18: $\widehat{n}_{opt} = $ minimum cost node at leaves
19: $\widehat{e}_{opt} = $ edge in tree level 1 leading to $\widehat{n}$
20: **return** $\widehat{e}_{opt}$

---

## 5. Migration to a Parallel Environment

In this section we describe our choices to realize an efficient many-core version of the algorithm presented above. They are mainly oriented to obtain a highly efficient tool able to run on an embedded system with constrained hardware resources like the ones available on modern vehicles.

### 5.1. High Level Tool Structure

CUDA programming adopts a SPMD (single-program, multiple-data) parallel programming style, and we design the algorithm to build the tree on a level by level basis.

As described in Figure 3, trajectories are organized as a tree of height $H$ and degree $D$. This tree includes all physically feasible paths taken into consideration. Among them, the best one is finally extracted based on a specific cost function. Algorithm 1 calls functions DRAWSAMPLE and EXPAND once for every tree edge. This meas that the algorithm performs several basic steps equal to:

$$1 + D + D^2 + D^3 + \ldots + D^H \quad = \quad \Sigma_{h=0}^{H} D^h \quad = \quad \frac{D^{H+1} - 1}{D - 1}$$

formulation easily derived by using the geometric progression.

In a highly parallel environment, it is somehow immediate to organize tree construction on a level-by-level basis using one thread to generate each single parent-to-child edge (trajectory). As for each tree level $i$, $D^i$ calls to functions DRAWSAMPLE and EXPAND will be made in parallel, the concurrent algorithm will be bounded by $H$ basic steps. Obviously, in a many-thread environment one of the main issues is how threads are synchronized and how they exchange information among them. Following this idea, Figure 4 revisits Figure 3 to show how the logic relationship among threads and how the overall data structure is organized. Figure 4 shows the data array $T^h$ used at each level.

From the logical point of view, each $T^h$ is a texture array, containing $D^h$ cells for each tree level $h$. From the implementation point of view, all $T^h$ textures are organized as a 2D matrix to exploit the 2D caching of the GPU. Threads refer to them using a 2D matrix index notation whereas in our logical explanation we often refer to a single index access. With this approach each tree level represents a set of parents for the next level and a set of children for the previous one. Data array at level $i$ is used to save the appropriate data to pass from threads at level $i$ to threads at level $i + 1$. Each array length is

equal to the number of tree nodes at that specific tree level. The array size grows exponentially, being 1, $D$, $D^2$, $D^3$, ..., $D^H$. The following relations hold for parent and children indexes: a node $i$ at a level $j$ has a parent from level $j - 1$ and $D$ children at level $j + 1$. The correspondence among those node indexes is the following:

$$PARENT(n_{i,h}) = n_{\lfloor \frac{i}{D} \rfloor, h-1} \qquad 0 < h \leq H$$
$$CHILD_j(n_{i,h}) = n_{[(i \cdot D) + j], h+1} \qquad 0 \leq h < H, \ 0 \leq j < D$$

Notice that this relationship is also a key aspect within the CUDA environment as each thread, given its position within the block-grid, has to save the computed data for its $D$ children.

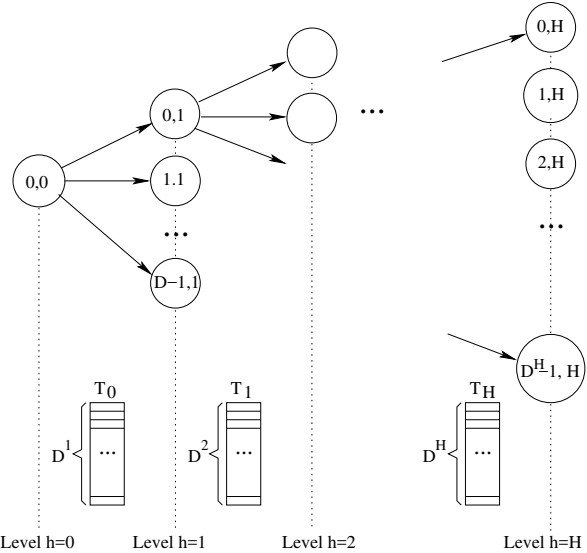

**Figure 4.** Tree, reserved memory to communicate information between layers, and mathematical dependency among tree nodes (i.e., concurrent threads). The tree has a degree equal to $D$ and a height equal to $H$. At each level $h$, we use a texture array $T^h$ containing $D^h$ elements, and implemented as a 2D matrix to exploit 2D caching strategies.

## 5.2. Data Structures and GPU Memory

Maps are built by the data fusion module which is external to the path planner. We consider map creation, manipulation, and transfer somehow outside the scope of the present paper. Anyhow, we need to keep into account all memory and CPU costs to perform such operations to properly synchronize our threads. As the data fusion module does not share any memory with the path planner, a critical aspect of the system is how to make the path planner communicate with the outside world. If we suppose that the data fusion module organizes its data as grid maps, those maps have to be transferred within the path planner and updated frequently, as a high refresh rate guarantees a better precision and a more dynamical behavior with respect to obstacles. This in turn also means to have large memory transfer costs and high memory occupation. For the above reasons, we decide to use a *surface* as a Read/Write data structure exploiting caching to optimize accesses. Working on surface memories has several performance benefits compared to using global memory.

In our implementation maps are accessed to evaluate the quality of our path, i.e., the cost of each position along the path. As we must represent the environment around the car dynamically changing along time, we use a different map for each tree level expansion.

In our framework, maps are represented as an image with a resolution of $(1000 \times 1000)$ pixels. Each pixel is described by a `float` value. Map resolution is 0.50 m meaning that a space of 500 m can be represented. Considering the resolution power of our sensors (see Section 6) maps are more than sufficient to represent the neighboring area.

We use GPU textures to store maps efficiently. As in texture memories each pixel is represented using the 4 RGBA channels (R, G, B and A), i.e., it is represented on 4 floating point values, we simultaneously represent 4 maps on a single texture by compressing 4 pixel floating-point representations into a unique RGBA field. As a consequence, a single texture in sufficient to encode all information required by an expansion tree with height $H \leq 3$. For tree with $H > 3$, one possibility would be to use more than one texture to represent the required number of maps. However, we experimentally noticed that for expansion trees higher than $H = 3$, all estimated maps become so approximated (This approximation is due to different reasons, such as the sensor inaccuracy and the erratic behavior of many objects present in the scene, e.g., pedestrians) that they loose their meaning. For that reason, when we analyze the space for a number of tree level higher than 3, we re-use the map for $h = 3$ for all higher levels.

In addition to occupancy grid maps the data fusion module sends to the path planner a mathematical path description together with the associated Voronoi diagram (In mathematics, a Voronoi diagram is a partitioning of a plane into regions based on distance to points in a specific subset of the plane. That set of points (called seeds) is specified beforehand, and for each seed there is a corresponding region consisting of all points closer to that seed to any other. These regions are called Voronoi cells). The drawing sample procedure, in both CPU and GPU versions, uses the Voronoi diagram for the generation of reference points. The procedure is the following one. When the car is not on the path and it is physically impossible to select a goal on the path, the algorithm selects a feasible goal and then it approximates such a goal with the closest point on the path. Finding the closest point on the path to a given goal point (in all steps) would be really time consuming. For that reason, we use Voronoi diagrams creating "Voronoi cells" including the set of points closer to the desired "Voronoi seeds" (i.e., the given goals).

The data structure containing the Voronoi diagram is an RGBA texture of $(1000 \times 1000)$ cells containing one `short` type. In the case of GPU implementation this diagram has to be transferred from host to a surface in the device memory. This process is expensive.

The basic data item necessary to compute a new tree edge includes the LVS and the steering angle, as well as additional information related to the current node cost. Each one of those data items requires 4 real values, i.e., 4 objects of type `float4` in CUDA. In our application all kernels share a common data structure. This can be stored on the global memory but with performance penalties.

The data structure is pre-allocated and overwritten at each execution of the planning cycle, and, as introduced in Figure 4, it can be seen as a set of layers, one for each tree level. Each layer stores all shown data, whose number depends on the current level.

## 5.3. High Level Algorithm

Our concurrent version of Algorithm 1 replaces the main iterations at lines 7 and 9 with concurrent thread computations. The overall work-load is then naturally partitioned in three conceptually independent tasks. Each of these tasks is implemented by a separate CUDA kernel:

- The DRAWSAMPLEKERNEL function computes reference nodes and speeds. This kernel, starting from source nodes, computes reference nodes for the second kernel and stores them in the surface memory.
- The EXPANDKERNEL function generates trajectories. This second kernel reads from the surface pair of source and reference nodes, and it computes the closest node according to the vehicle kinematic model. Results are made available in the surface memory for the next execution of the other two kernels. A cost is stored for each node, keeping into account the node's parent cost.
- The COMPUTECOSTKERNEL function computes the final path. This kernel efficiently identifies the minimum cost node, and the best physically feasible path to the root.

Algorithm 2 presents our concurrent version of Algorithm 1.

Lines from 1 to 4 follow the sequential algorithm. On line 5, function MEM2SURF stores the required data from the CPU memory to the GPU surface memory. Within the main loop (line 6),

the application runs twice $D^h$ groups containing $D$ threads for a total of $2 \cdot D^{h+1}$ threads per cycle. The first set of $D^{h+1}$ threads runs through the DRAWSAMPLEKERNEL function, and the second set runs through the EXPANDKERNEL code. Those kernels are run in sequence, and are logically kept separated. This is because we want to keep both kernels simple enough, and to avoid branches such that the SPMD programming style is preserved.

---

**Algorithm 2** Highly-parallel (i.e., many-core) top-level path planner algorithm.

      CONCURRENT PLANNING CYCLE
  1: $N = \varnothing$, $E = \varnothing$
  2: $n = (LVS, T_0)$, $N = N \cup \{n\}$
  3: $(\widehat{n}, \widehat{e}) = $ SIMULATE $(n, f, g, T_{cycle}, u_0)$
  4: $N = \cup \{\widehat{n}\}$, $E = E \cup \{\widehat{e}\}$
  5: MEM2SURF $(surf, \widehat{n})$
  6: **for** $h = 0$ to $H - 1$ **do**
  7:     DRAWSAMPLEKERNEL $<D^h,D>$ $(surf, VoronoiMap, Path)$
  8:     EXPANDKERNEL $<D^h,D>$ $(n, f, g, d_{ref}, v_{ref}, T_{sim}, surf, gridMap)$
  9: **end for**
10: COMPUTECOSTKERNEL $<D^H,1>$ $(surf)$
11: $\widehat{n}_{opt} = $ minimum cost node at leaves
12: $\widehat{e}_{opt} = $ edge in tree level 1 leading to $\widehat{n}$
13: **return** $\widehat{e}_{opt}$

---

Please remind that to avoid excessively long waiting times, all threads within the same kernel should execute in close *time proximity* with each other. Anyhow, the CUDA run-time system satisfies this constraint by assigning execution resources to all threads in a CUDA block as a unit, that is, when a thread of a block is assigned to an execution resource, all other threads in the same block are also assigned to the same resource. This ensures time proximity of all threads in a block and prevents excessive waiting time during barrier synchronization.

Moreover, to enforce regularity to all computations performed along each tree path, we avoid tree pruning even when certain edges are not useful anymore (for example, when an obstacle is too close to the computed path).

Another main issue of the algorithm is thread parallelism. For each tree layer $h$, with $h$ starting from 0, we have $D^{h+1}$ concurrent threads. As we suppose to set $D = 6$, we will have $6^{h+1}$ threads running in parallel. As in our experiments, we used a GPU with 1664 cores, its parallel capability will saturate with $h \geq 5$. As we build the tree in a breadth-first way, this also means that we obtain the maximum parallelism in the last level whereas the parallelism is quite low during previous levels. To further increase the parallelism obtained, we have schemes in which the degree $D$ is trimmed during the process, being larger during lower tree levels and higher for higher tree levels.

The first two kernels are executed sequentially, one after the other, $H$ times. Only when the tree is complete, the third kernel (see description in Section 5.6) is launched. To efficiently run those two kernels, we organize our data structure as previously described, i.e., as a sequence of arrays containing nodes belonging to the tree and stored within surface memories. Working on surface memories has several performance benefits compared to using global memory.

Threads executed by different kernels share data and logically relate to each other using the same data structure. Figure 5 illustrates the communication between threads. In this picture, time flies from left to right, and we highlight how working threads at level $h = 0$ generate information for the working threads at level $h = 1$. We suppose $D = 6$.

At the very beginning, the algorithm concentrates only on the tree root representing the initial vehicle position. This position is stored in a single data record named $src_1$. The first kernel DRAWSAMPLEKERNEL evaluates the first set of 6 target points. To perform this step, as represented in Figure 6a, the initial vehicle position is projected along the current path, and then orthogonal to the desired path, to find $D$ (6 in the picture) projected points.

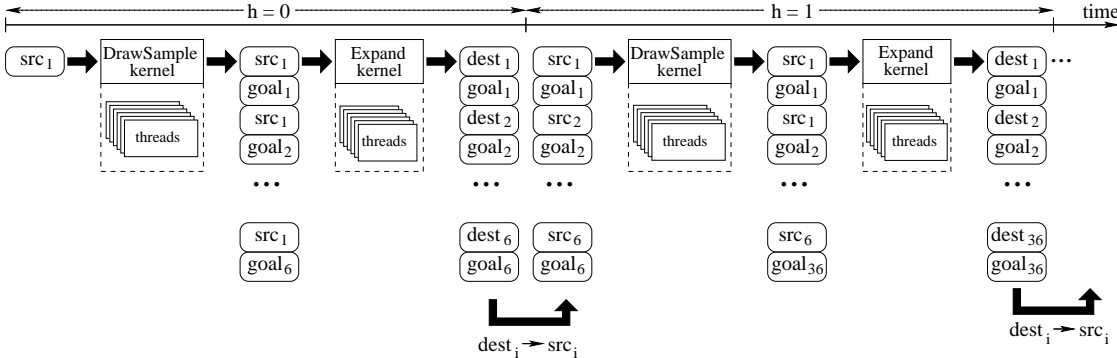

**Figure 5.** Memory organization for thread communication concentrates on a tree of degree (*D*) equal to 2. For $h = 0$, the DRAWSAMPLEKERNEL kernel generates $D = 6$ goal destinations $goal_i$. The EXPANDKERNEL kernel tries to reach these goals, and it generates 6 destinations $dest_i$, as close as possible to the corresponding goal. For $h = 1$ all operations are repeated starting from 6 source positions (the 6 destinations reached at the previous iteration).

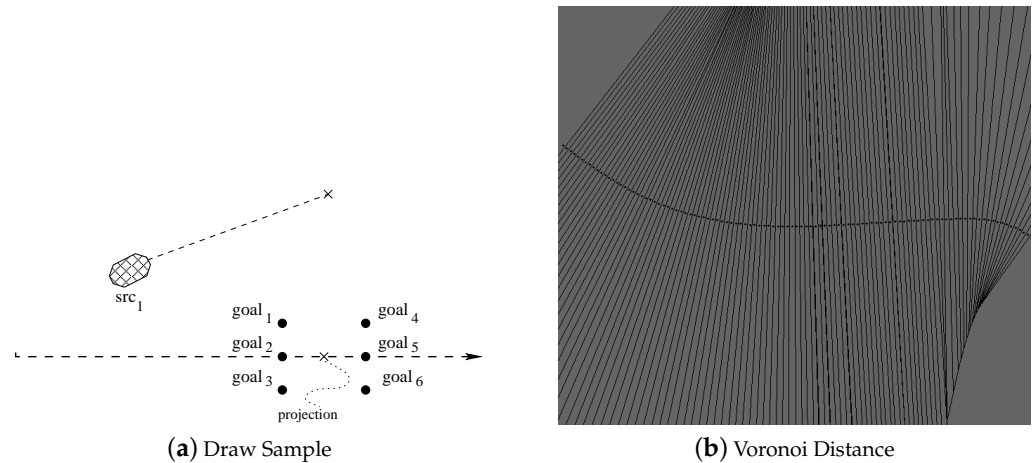

(**a**) Draw Sample                    (**b**) Voronoi Distance

**Figure 6.** (**a**) shows an example of how procedure DRAWSAMPLEKERNEL generates the goal samples already analyzed in Figure 5. The Voronoi map (**b**) shows all points with the minimum distance for any point along the desired reference path (i.e., the points on the normals to the path itself).

Notice that as the planner is running over and over again to compute paths in close proximity of one another, orthogonal projections must be computed and recomputed for many points close to the vehicle. To avoid those re-computations and to make DRAWSAMPLEKERNEL more time efficient, all orthogonal projections for dense points around the vehicle are computed during the map generation for all maps points at fixed time intervals. This is allowed by a proper use of Vonoroi maps. Figure 6b is a logical representation of the Voronoi map, where for each point *p*, the segment passing from *p* and orthogonal to the path identifies the point on the path closest to *p*. Each Voronoi map is stored into a texture of size equal to $(1000 \times 1000)$ pixels. Within the Voronoi map, each pixel is represented with 4 float values stored as an RGBA information. The first float indicates the index of the orthogonal projection path point stored within the texture path. All other float values store the angle of the tangent to the path and its orthogonal direction. These data are used by the kernel to compute the goals. Each map includes the GPS coordinates stored within the first top-left pixel. This pixel also specifies the pixel density within the map, i.e., the real distance between two points. This density is a function of the vehicle speed at the moment the map is generated by the global planner. Each thread reads this information and it is then able to compute the lookahead position on the map. Please note that as each map may serve several path planning cycles, it has to be large enough to include all lookahead positions computed within the next few cycles.

Figure 5 shows how kernel DRAWSAMPLEKERNEL modifies the memory structure to set-up all required data to run kernel EXPAND. Record $src_1$ is expanded into 6 record pairs $src_i$-$goal_i$. For each couple, $goal_i$ is the target positions (the ones represented in Figure 6a), and $src_1$ is the same source, common to all trajectories that must be computed by kernel EXPANDKERNEL. When the second kernel EXPANDKERNEL runs, each $src_1$ record is replaced by the simulated destination position $dest_i$. In this way, those destinations will be considered as new sources during the next algorithm iteration. This step is also represented by Figure 7a, whereas Figure 7b represents a mock grid map and a possible vehicle trajectory. Notice that at the end of our tree construction $dest_i$ are the final vehicle positions. Then each path from $src_1$ to a $dest_i$ implicitly includes the set of commands (generated by EXPANDKERNEL) that are necessary to reach a destination that, at least theoretically, should coincide with the corresponding $goal_i$ at $T_{lookahead}$. Unfortunately, the algorithm just approximates desired paths. As a consequence, we store pairs $dest_i$-$goal_i$ at the end of step $h = 0$. When the second iteration of the loop at line 6 (the one with $h = 1$) of Figure 5 starts the 6 destination points $dest_i$ found during the previous iterations become source points $src_i$. The second iteration will proceed as the previous one, but starting with $D = 6$ points procedure DRAWSAMPLEKERNEL will generate $D^2 = 36$ points, and procedure EXPANDKERNEL will target them.

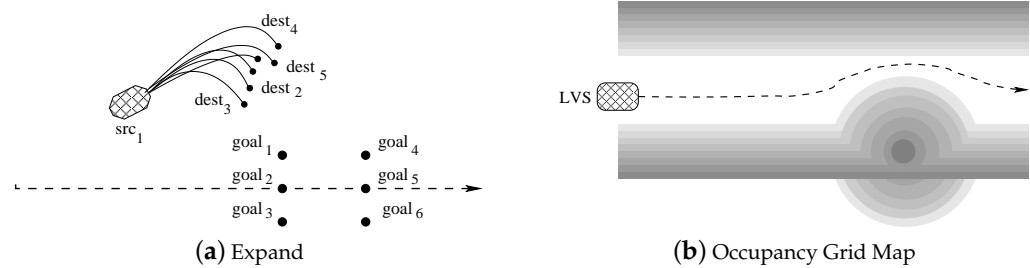

(**a**) Expand                    (**b**) Occupancy Grid Map

**Figure 7.** (**a**) shows an example of how procedure EXPANDKERNEL computes destination points by expanding the recursion tree of one single level. (**b**) shows all generated points (the entire path) within the current occupancy grid.

### 5.4. Function DRAWSAMPLE: Finding Target Points

Function DRAWSAMPLEKERNEL prepares the necessary data for kernel EXPAND at the beginning of each cycle or tree level.

As represented in Figure 5 by dotted boxes, at the tree level $i$, DRAWSAMPLEKERNEL runs a thread group of size $D$ for each source node. Each group works on the same source node to generate $D$ destination nodes. This source node is directly read from the surface memory. This operation is represented in line 1 of Algorithm 3, where the node is addressed using the group index $blockIdx.x$.

Function DRAWSAMPLEGPU is a modified version of function DRAWSAMPLE, introduced in Algorithm 1. Starting from the unique single reference node collected in $src$, each thread $threadIdx.x$ within DRAWSAMPLEGPU performs the following steps:

- It computes the lookahead position, i.e., the position of the vehicle without any new command, at future time $T_{lookahead}$.
- Starting from the lookahead position within the Voronoi map, it computes the orthogonal projection onto the path.
- Given the path, the planner (node or) path splitting policy, and its $threadIdx.x$ index, it generates a new goal $goal_i$.

---

**Algorithm 3** kernel1: DRAWSAMPLE

---

    DRAWSAMPLEKERNEL
      1: *src* = SURFREADSRC (*blockIdx.x*)
      2: *goal* = DRAWSAMPLEGPU (*src, threadIdx.x*)
      3: SURFACEWRITEREF (*src, goal, blockIdx.x, threadIdx.x*)

---

*5.5. Function* EXPANDKERNEL: *Computing Path to Target Nodes*

Following Section 5.1, at tree level *i* a thread group of size *D* is launched for each pair (*src, goal*). Each thread executes the kernel described in Algorithm 4. For each group the source node is read from the surface, whereas the reference node for each thread *threadIdx.x* is the one computed by the first kernel.

Function EXPAND is the same of Algorithm 1. The output of EXPAND is the reached point (*dest*) using functions *f* and *g* and represents the source node for the following execution of Algorithm 3.

Line 3 is in charge of computing the cost of a generated node and the corresponding edge. Function COMPUTEWEIGHTANDCOLLISION computes node costs. Each node has a cost that derives from its parent node cost and its position within the grid map. For this kernel, grid maps have a structure similar to the one described in Section 5.1. Nevertheless, these grid maps have to serve function EXPAND for *H* consecutive calls, corresponding to *H* tree levels of the expansion tree. Each map has thus to foresee all object movements within $T_{lookahead}/H$ time unit. As we represent each pixel with 4 RGBA float values, we are able to represent grid maps up to 4 unit of time. If the $H > 4$, the last map is reused by all kernel calls following the fourth one. Notice that this choice does not invalidate results, as in any case the last map is the one in which the trajectory is foreseen less precisely, and thus reusing it does not entail larger errors.

We use *exponential averaging* to compute new costs, given higher weights to more accurate estimated positions, i.e., nodes closer to the root. For a new node at level *h* the cost is computed based on the cost of all nodes along the path leading to this node from the root:

$$cost_h \quad = \quad cost_0 \cdot \alpha^0 + cost_1 \cdot \alpha^1 + \ldots + cost_{h-1} \cdot \alpha^{h-1}$$

where $\alpha \in [0, 1[$, $cost_0$ is the coefficient for the closest estimate (after $1 \cdot (T_{lookahead}/H)$ time units), and $cost_{h-1}$ is the farther estimate (after $h \cdot (T_{lookahead}/H)$ time units). The destination node is marked as unfeasible when required. This essentially depends on how the grid maps are generated and on how the trajectory is placed on such a map.

Function SURFACEWRITETREE writes nodes on the surface to set up all required information for the next iteration of the main cycle of Figure 2. Each thread works on one tree layer overwriting old information (all source and goal points written by function EXPAND) with source and destination points.

---

**Algorithm 4** Kernel2: EXPAND

---

    EXPANDKERNEL
      1: $(n_{src}, n_{dest})$ = SURFREAD (*blockIdx.x*)
      2: $(\widehat{n}, \widehat{e})$ = EXPAND $(n_{src}, f, g, n_{dest}, T_{sim})$
      3: COMPUTEWEIGHTANDCOLLISION $(\widehat{n}, \widehat{e})$
      4: SURFACEWRITETREE $(\widehat{n}, \widehat{e}, blockIdx.x, threadIdx.x)$

---

*5.6. Function* COMPUTECOSTKERNEL: *Selecting the Best Path*

Once all trajectories have been computed and their costs evaluated, tree leaves contain the cumulative cost of the entire path leading to them. The next step is to select the most promising trajectory, i.e., the one with the smallest cost. Finding a minimum entails a linear visit, but implementing a linear visit on a multi-core architecture can lead to several inefficiencies. As suggested by many other authors (see for examples Chen at al. [14] for considerations on many-thread sorting) we trade-off time-efficiency and accuracy. Figure 5 describes our bucket sort-inspired algorithm. It works as follows.

The cost function computes real cost values for each leaf. Let us suppose those values are included in a specific interval $[l, r]$. First of all, we divide this interval into $N$ classes. In this way, each class has a width equal to $(r - l)/N$. Then, we build a pseudo-histogram by inserting in each class all leaves with a costs belonging to the class interval. To populate the histogram, i.e., to insert each leaf in the proper class, function COMPUTECOSTKERNEL runs one thread for each tree leaf.

Each thread behaves like function POPULATEHISTOGRAM in Algorithm 5. Each thread is in charge of placing the leaf identifier into the corresponding histogram class. To do that, it gets the leaf identifier and the leaf cost from its leaf (lines 1 and 2). Given the leaf cost $leaf_{cost}$, line 3 computes the index of the bucket ($histogram_{index}$) the leaf belong to. As all threads work in parallel, we must guarantee a proper synchronization among them, such that only thread can modify a class at any given time. To do that, we use the atomic CUDA function ATOMIC_ADD (see line 4) to add the node identifier to the proper class bucket (properly initialized to zero). As the CUDA ATOMIC_ADD returns the original value for each addition, we always know whether the added value is the first one or not (line 5). In the first case, the thread leaves the bucket equal to the leaf identifier and then it terminates. Otherwise, it subtracts the same leaf identifier from the bucket (line 6) such that when all threads have terminated each class bucket stores only one identifier value, corresponding to the node placed in the bucket first.

Once all threads running function POPULATEHISTOGRAM have terminated, function COMPUTECOSTKERNEL runs one more kernel with a single thread. This thread performs a linear search in all buckets of the histogram looking for the leaf with the smallest cost, i.e., the one stored in the leftmost bucket.

Notice that in this case linear search is performed only on those classes that have no representative, as the algorithm stops on the first non-empty class. This makes our algorithm much faster than a standard linear search. Moreover, we select the number of classes $N$ as a function of maximum available number of threads available and the desired approximation. For example, if our costs belong to the interval $[0.0, 1.0]$, and we select $N = 1000$, we generate a histogram with 1000 classes, and we obtain a class width and an accuracy equal to 0.001.

As a last step, function COMPUTECOSTKERNEL returns the selected node plus the the entire path leading to it from the tree root.

---

**Algorithm 5** Histogram Computation.

---

POPULATEHISTOGRAM
1: $leaf_{id}$ = RETRIEVENODEINDEX()
2: $leaf_{cost}$ = RETRIEVENODECOST()
3: $histogram_{index} = \lfloor \frac{leaf cost}{r-l} \cdot N \rfloor - 1$
4: $oldVal$ = ATOMICADD(bucket[$histogram_{index}$], $leaf_{id}$)
5: **if** $oldVal \neq 0$ **then**
6:    ATOMICADD(bucket[$histogram_{index}$], $-leaf_{id}$)
7: **end if**

---

## 6. Experimental Analysis

As our goal has been to analyze the complexity and implications to re-implement an existing algorithmic-based and efficient strategy on a many-core embedded architecture. For this reason, to really compare apple-to-apple, the main goal of our experimental analysis is to compare our CPU implementation of the algorithm with the parallel and optimized one running on an embedded GPU. Explicitly comparing our GPU implementation with other ones would be meaningless, as it would eventually grade the quality of the original algorithm, we did not improve or modify at all, against other similar or dissimilar strategies.

Moreover, notice that the project has been developed under an industrial non-disclosure agreement between Politecnico di Torino and Magneti Marelli. For that reason, the software and the experimental results cannot be made publicly available.

Our environment follows the path planner structure represented in Figure 1. Our planner (fully implemented in CUDA language) works in close loop with a vehicle controller and a vehicle simulator acting as the external environment. To be as complete as possible, we compare our CPU implementation with our GPU one in terms of the quality of the paths gathered, algorithm scalability, and real-time response time. Quality is evaluated in terms of the metrics described in Section 6.2, scalability and response time in terms of wall-clock (elapsed) execution times and number of generated trajectories.

Our results have been collected by running our implementations on the following hardware devices:

- A CPU Intel Core i7-6700 HQ with 2.60 GHz and 8.00 GB of RAM memory.
- A GPGPU NVIDIA GEFORCE GTX 970 with 1664 Cores and 4.00 GB of RAM memory.

Section 6.1 describes our working scenarios and their relationship with the real world. Section 6.2 introduces our quality and efficiency evaluation metrics. Section 6.3 reports our work to trim the original algorithm to its best. Section 6.4 compares the CPU and the GPU versions of the planner in terms of computation times.

### 6.1. Operating Scenarios

Countries and organizations define driving cycles to assess the performance of vehicles in various ways, as for example fuel consumption and polluting emissions. This represents a standard in the automotive industry. For example, Figure 8 shows the New European Driving Cycle (NEDC) driving cycle (updated in 1997 for the last time) designed to assess the emission levels of car engines and fuel economy in passenger cars (which excludes light trucks and commercial vehicles).

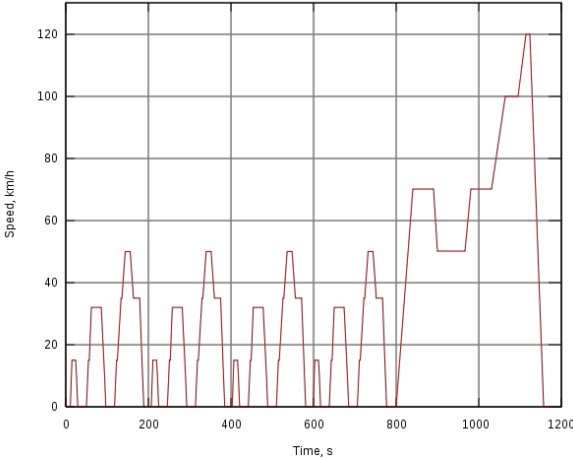

**Figure 8.** The New European Driving Cycle (NEDC): A driving cycle designed to assess the emission levels of car engines. It is also referred to as MVEG cycle (Motor Vehicle Emissions Group).

Unfortunately, such a standardized driving cycles have not been yet defined for evaluating a trajectory planner for autonomous navigation. Thus, the quality of local trajectory planner techniques can be proved only in user-defined test scenarios.

Operating scenarios range from parking areas, to urban roads, and to highways. We need to differentiate situations where no external structure could be extracted for guidance (such as driving off-road or parking in large-scale parking lots), from the one in which a reference path may be made available (such as urban roads, highways, and, in general, roads with some sort of lane identification). For a trajectory planner working with a reference path, a very common scenario is driving on a highway. In this case relevant benchmark tests are:

- Path following: On a straight path, a low curvature (A curvature is a measure of how quickly a tangent line turns on a) path ($<$0.0081/m), and a high curvature path ($>$0.0081/m).

- Lane change: With normal condition (a lateral acceleration of $2 \, \text{m/s}^2$), and with obstacle avoidance (a lateral acceleration up to $9.81 \, \text{m/s}^2$).

For example, Figure 9 shows a straight and a low curvature paths with the expansion tree drawn by our application.

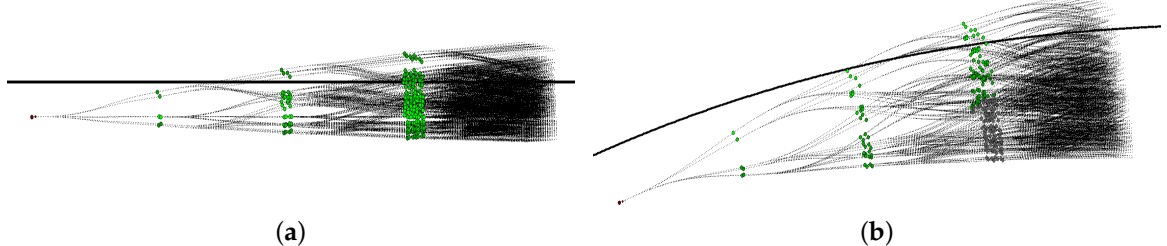

| (a) | (b) |

**Figure 9.**   Real paths and real expansion trees (logically described in Figure 3) for a straight path (**a**) and a curved path (**b**). The car is initially placed at the center of the lane and its target is to converge on the specified path (black line). To have an idea of the convergence speed, the lane is 6 m wide and the trajectory about 40 m long.

To reduce the number of cases we must present and discuss, we concentrate on a more complex combined evasive maneuver, more commonly knows as the *moose* or *elk* test. Forms of the test have been performed in Sweden since the 1970s, it has been standardized in ISO 3888-2, and it is usually performed to determine how well a certain vehicle evades a suddenly appearing obstacle. With a moose test, we simulate a sequence of path followings and of lane changes at the same time. Moreover, this test can also be seen as a vehicle overtaking or as a vehicle obstacle avoidance. The first obstacle (the closest one) is on the same lane the vehicle is moving on, whereas the second obstacle is on the fast lane. The distance between the two obstacles is set to 75 m. The reference path is at $y = 0$ m, and the vehicle starts simulation at $y = -3$ m at the center of the right lane (as in Figure 9). Grid maps limit the road from $y = -6$ m to $y = +6$ m.

For our analysis we selected a highway with the following characteristics: Roadway width 6 m, minimum curvature radius 340 m, and vehicle speeds spanning between 13 m/s (46.8 Km/h) and 36 m/s (129.6 Km/h). In normal driving conditions, accelerations vary in the range $[-2 \, \text{m/s}^2, +2 \, \text{m/s}^2]$. The vehicle width is set to 2 m. Given the vehicle and roadway widths, lateral offset is at most 0.8 m. All simulations are conducted considering a sensor configuration capable of identifying obstacles in a 140 m radius with a 1.5 s delay to process an obstacle from its appearance to its recognition. Sensor data are manipulated by a data fusion module which creates all grid maps and all other information required by the planner. Maneuver are accomplished only using a kinematic model compatible with the vehicle and the environment parameters. Suitable speeds for the experiments are computed using motion equations and the available vehicle models. Moreover, in our setting the vehicle control system sends commands to the actuators at fixed frequency rate, varying from 50 Hz to 100 Hz. As the controller produces commands based on the trajectories generated by the path planner, the planner working time is bounded by the controller frequency.

*6.2. Evaluation Metrics*

We evaluate our implementations using the following four metrics.

The *Starting Distance* (*SD*) is the distance between the space point at which we start the maneuver and the obstacle. It indicates how fast the algorithm is to react to a stimulus. Usually, the higher is the starting distance, the safer is the maneuver.

The second metric is the *Root Mean Square Error* (*RMSE*), which can be expressed as:

$$RMSE \quad = \quad \sqrt{\frac{\sum_{i=0}^{N} \sqrt{(x_{gi} - x_{pi})^2 + (y_{gi} - y_{pi})^2}}{N}}$$

where $(x_{gi}, y_{gi})$ is the i-th path planner generated point, $(x_{pi}, y_{pi})$ is the i-th point on the path to follow, and $N$ is the number of generated points. *RMSE* essentially measures the root of the mean square distance between the computed trajectory and the desired one. Usually, *RMSE* as to be as small as possible.

The third metric is the *Minimum Obstacle Distance* (*MOD*). It shows the minimum distance reached from the obstacle, and it is a measure on how safely the path has been placed on the scene. From the one hand, its value has to be as large as possible even if it cannot exceed infrastructure size (e.g., it is unsafe to perform an overtaking maneuver maintaining a distance to the overtaken vehicle larger than the lane width). On the other hand, it strongly depends on driving style (relaxed, fuel-efficient, sportive, etc.). This also depends on how grid maps have been designs, as a shorter gradient between white (admitted) and dark (non-admitted) areas implies more sporty driving styles and vice-versa.

Finally, we compare the CPU against the GPU implementation in terms of wall-clock times. The wall-clock time is the time necessary to a (mono-thread or multi-thread) process to complete its job on a new problem, i.e., the difference between the time at which the problem is completely handled and the time at which this task started. For this reason, wall-clock time is also known as *elapsed time*. This is an important measure as the faster the computation, the deeper and broader we can span the space around the vehicle, and the higher the frequency at which we can update a path. This is particularly important in emergency driving conditions. Emergency driving conditions happen when the car is forced to perform a very sharp maneuver such as the one required to avoid an unexpected obstacle. In this cases the car could reach accelerations higher then 2 m/s$^2$ in absolute value. Emergency driving conditions are somehow beyond the scope of this paper, but we use them to compare the efficiency of our GPU algorithm to the original CPU-based one.

*6.3. Original Algorithm Parameter Setting*

Algorithm 1 is influenced by several parameters. Among those, a few ones, such as the tree degree $D$ and the tree height $H$, have a larger impact on the tree structure, the degree of parallelism, and the required memory. Other ones, such as $T_{sim}$ and $T_{lookahead}$ have a large impact on the path planner timing and accuracy.

The degree of the tree $D$ has a high impact on the path planner as the higher the number of trajectories the higher the covered region around the vehicle. The value of $D$ is selected based on a global *splitting* policy. This policy in turn is a function of the maneuver the global planner decided to undertake and of the environment around the vehicle. As described in Section 4.2, $D = |O| \times |V|$, where $O$ is a discrete set of lateral offsets, and $V$ is a set of longitudinal vehicle velocities. If the global planner decides to enforce a rapid velocity changes (i.e., extreme acceleration or deceleration conditions) it will enforce large value of $|V|$. On the contrary, if it decides to obtain abrupt direction changes (i.e., parking maneuver or obstacle avoidance) a large value of $|O|$ will be set.

For the sake of space, we do not present any experimental evidence on $D$ in this section, and we present results in Section 6.4, showing that we often obtain the best trade-off with $D = 6$. To obtain this value, we adopt a splitting policy in which $|O| = 3$, such that the tree spans the entire lane from border to border (left-border, center, and right-border), and $|V| = 2$, with two speeds that are equal to the current one $\pm\Delta$. On the contrary, Figures 10–12 show our experimental results to set the value of $H$, $T_{sim}$, and $T_{lookahead}$. As we will see, $H$ has high impact on the level of parallelism it is possible to obtain with the concurrent application. At the same time, $T_{sim}$ has a consistent impact on the elaboration time. Finally, $T_{lookahead}$ has a great influence on the reaction to obstacles and, as a consequence, on the computed trajectory.

Figures 10e, 11e and 12e concentrate on the *moose* or *elk* test. The two obstacles are represented by black rectangles, and as described in Section 6.1 the first obstacle is on the same lane, whereas the second one is on the fast lane. The distance between the two obstacles is 75 m.

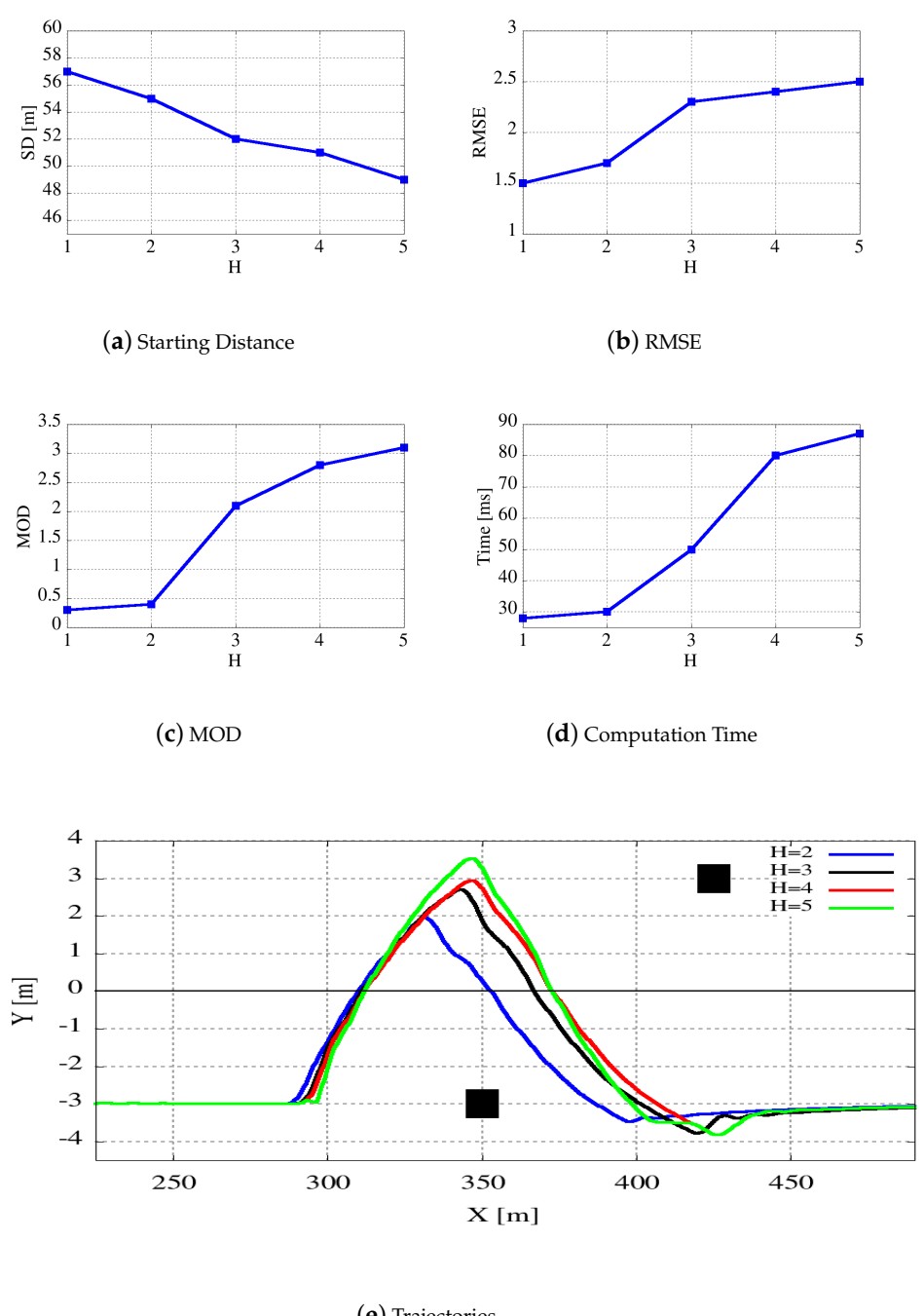

(**a**) Starting Distance

(**b**) RMSE

(**c**) MOD

(**d**) Computation Time

(**e**) Trajectories

**Figure 10.** Parameters and paths evaluation for an overtaking maneuver as a function of the height of the tree *H*, varying from 2 to 4. Graphs plot the starting distance (**a**), the root mean square error (**b**), the minimum obstacle distance (**c**), and the computation time (**d**). (**e**) shows the reference path and the final trajectories obtained with the different parameters. The car speed is fixed at 25 m/s (90 km/h).

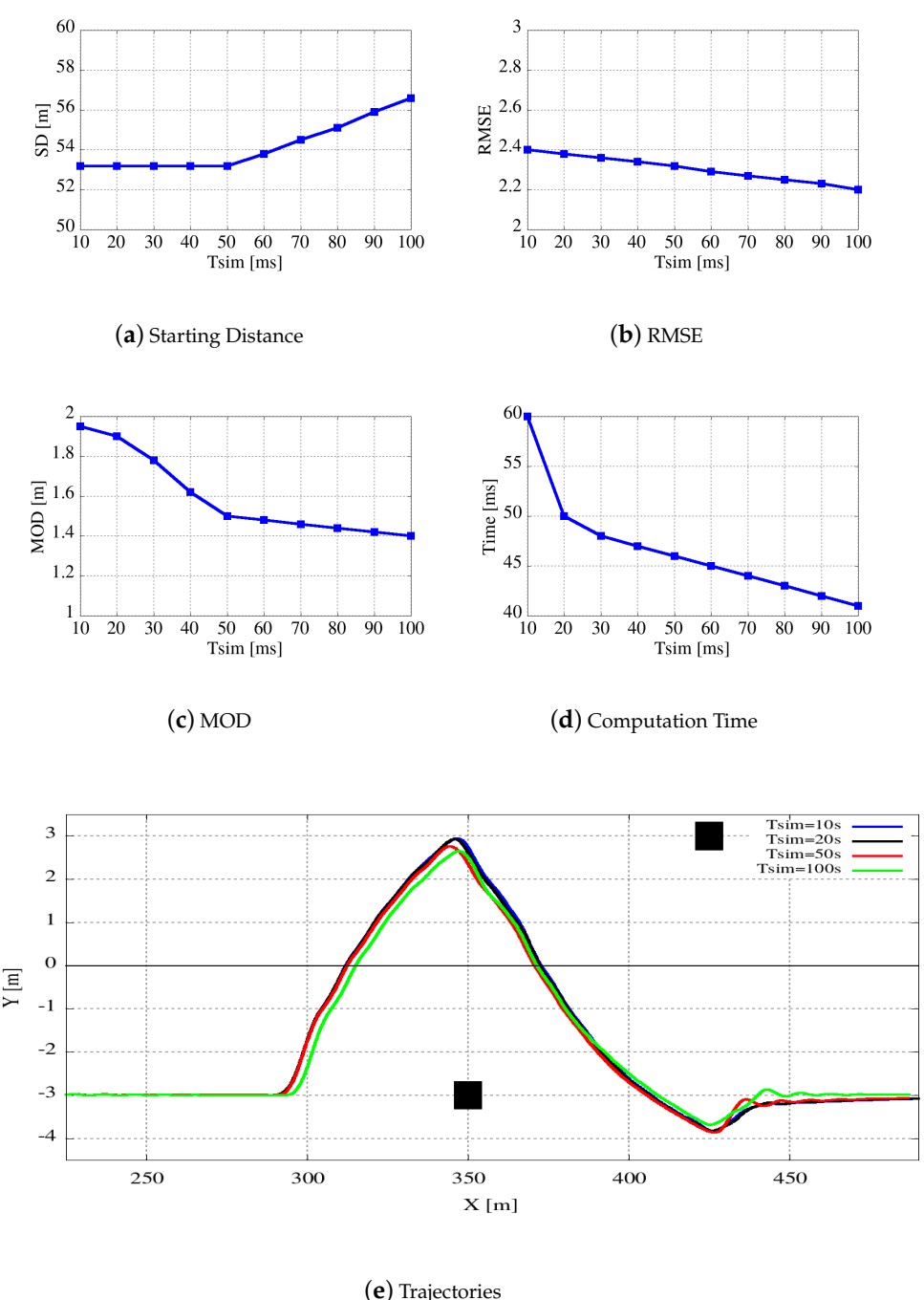

(**a**) Starting Distance

(**b**) RMSE

(**c**) MOD

(**d**) Computation Time

(**e**) Trajectories

**Figure 11.** Parameters and paths evaluation for an overtaking maneuver as a function of the height of the $T_{sim}$ parameter varying from 10 ms to 100 ms. Graphs plot the starting distance (**a**), the root mean square error (**b**), the minimum obstacle distance (**c**), and the computation time (**d**). (**e**) shows the reference path and the final trajectories obtained with the different parameters. The car speed is fixed at 25 m/s (90 km/h).

Each figure is essentially composed by 5 pictures. The first three small graphs plot the starting distance, the root mean square error, and the minimum obstacle distance from the obstacle as a function of one of the parameters $H$, $T_{sim}$, or $T_{lookahead}$. The fourth small graph plots the computation time, i.e., the wall-clock or elapsed time to compute the entire tree and to select the best trajectory. In this first set of experiments all times are evaluated on the CPU version of our application, used a base line for our tool. The CPU to GPU comparison is reported in Section 6.4. The largest graph, at the bottom,

shows the reference path for the maneuver and all paths generated using the tool with a few different settings. The red dot represents the overtaken vehicle.

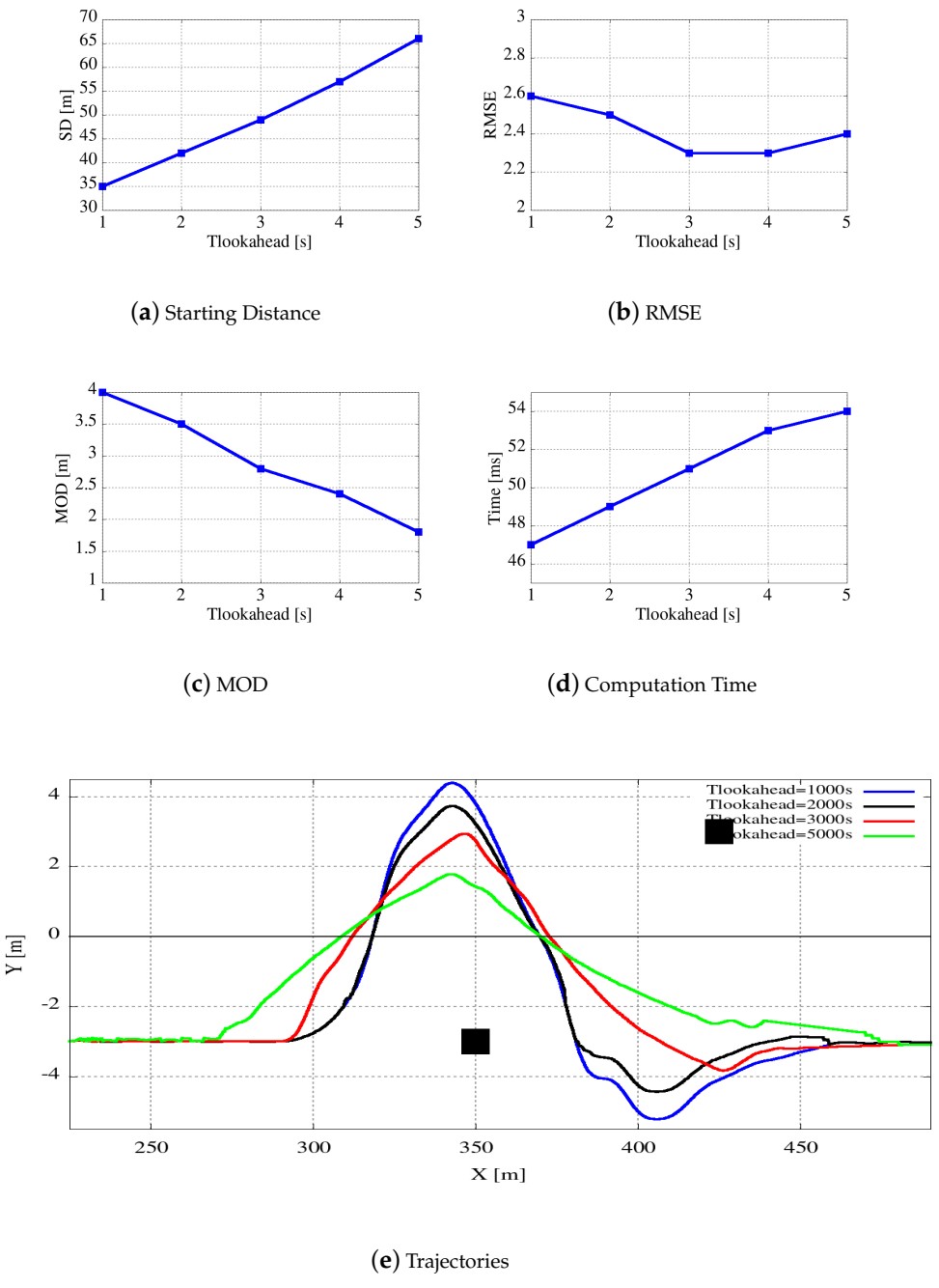

(**a**) Starting Distance

(**b**) RMSE

(**c**) MOD

(**d**) Computation Time

(**e**) Trajectories

**Figure 12.** Parameters and paths evaluation for an overtaking maneuver as a function of the height of the tree $T_{lookahead}$ parameter varying from 1 to 5 s. Graphs plot the starting distance (**a**), the root mean square error (**b**), the minimum obstacle distance (**c**), and the computation time (**d**). (**e**) shows the reference path and the final trajectories obtained with the different parameters. The car speed is fixed at 25 m/s (90 km/h).

Figure 10c presents the same plots of Figure 10a,b as a function on the height of the tree $H$, varying from 1 to 5. Higher values of the tree height reduce the starting distance and increase the *RMSE* metric, but reduce the minimum obstacle distance. This trend may somehow be justified by the fact that the most influencing tree section for the planning quality is the first level. When changing the tree height,

trajectory lengths remain the same but edges are shorter. From a computational point of view more layers exponentially correspond to more nodes, and that implies a growth of the computation time as shown in Figure 10c. At the same time, more nodes imply a higher parallelism on the GPU version of the algorithm. For that reason a value of tree height equal to 4 is preferred to reach the right balance among several parameters.

Figure 11 shows the behavior of the algorithm as a function of $T_{sim}$. The plots show that $T_{sim}$ does not influence too much the computed trajectory within a wide range of values as Figure 11a–c show modest variations. While Figure 11a,b would suggest values close to 100 for $T_{sim}$, Figure 11c suggests values close to 20. Please note that, as previously stated, $T_{sim}$ has a consistent influence on the elapsed time because varying $T_{sim}$ from 10 to 20 ms reduces computation times from 60 to 50 ms. Keeping into account the computational efforts required for the generation of the trajectories, plotted in Figure 11d, we select $T_{sim}$ = 20 ms for all our subsequent experiments.

Figure 12 presents the same plots of Figure 11 as a function of $T_{lookahead}$, varying from 1000 ms to 5000 ms. It can be noticed that the lookahead time has an impact on the computed trajectories much larger than $T_{sim}$. Figure 12a shows that a higher lookahead time corresponds to an early obstacle detection. On the other hand, Figure 12c shows that an early vehicle detection together with a deeper knowledge of the future path decreases the minimum obstacle distance from the overtaken obstacle. This means that the lookahead time influences vehicle behavior in terms of driving style, which is also related to the comfort perceived by passengers. Finally, Figure 12b shows that the root mean square error is not influenced that much by $T_{lookahead}$. A quite conservative value for $T_{lookahead}$ can be around 3000 ms, as with this value the resulting trajectories run not too close but not too far from the obstacle. As a summarizing remark, notice that higher $T_{lookahead}$ values are more suited to highway routes, where starting distance may have a higher priority with respect to minimum obstacle distances, On the contrary, smaller $T_{lookahead}$ values are better for parking maneuvers where speed is drastically reduced and obstacles are usually motionless.

### 6.4. Time Comparison

This section is devoted to a numerical comparison in term of performance between the CPU and the GPU version of the presented path planner.

Table 1 compares computation times for the CPU and the GPU version as a function of tree degree $D$ and tree height $H$. Our splitting policy generates the set of points represented in Figure 13.

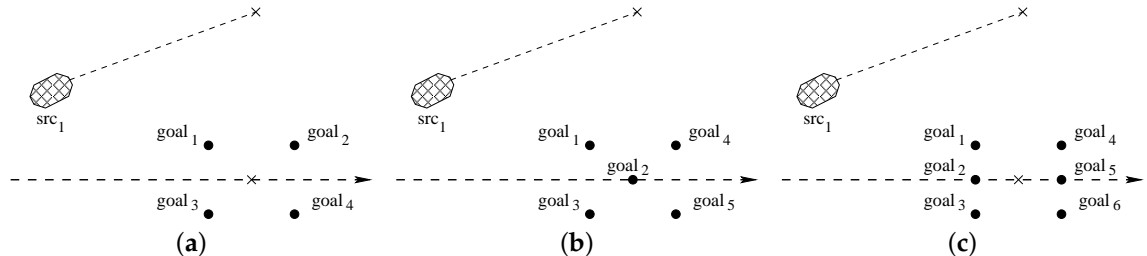

**Figure 13.** A graphical representation for the $D$ points selected by kernel DRAWSAMPLEKERNEL based on our splitting policy. The initial direction of the car must be modified to converge toward $D = 4$ (**a**), $D = 5$ (**b**), or $D = 6$ (**c**) different goals, respectively. The picture shows how goals are selected based on the initial curve projection on the desired path.

Notice that reported times are somehow independent from the path followed and maneuver computed, as the amount of computation performed does not vary with the trajectory. Column $D$ indicates the tree degree, and $H$ the tree height. Given those two values we generate a defined number of trajectories (column # TRAJECTORIES). M indicates the memory occupancy (in $k$ Bytes). For the CPU implementation each node requires 10 float values, each one requiring 4 bytes of memory. For the GPU implementation each node requires 16 float values, stored as 4 pixel textures RGBA,

each one requiring 4 bytes of memory. The textures used to store trees are square matrices of size $L$, with $L = 32, 55, 79, 124, 147, 280$ in the reported configurations. Computation times (in $m$ seconds) follow in the next columns. $T_1$ is the time required to build the tree, whereas $T_2$ is the time required to run COMPUTECOST, and Tot is the total times, i.e., the sum of those to values.

Data show that our CPU version is somehow comparable with the one by Schwesinger et al. [10] albeit different hardware architectures have been adopted, and possibly several implementation details may differ.

The GPU implementation outperforms the CPU variant. The performance gap increases with the size of the exploration tree. The GPU implementation is able to generate more trajectories than the CPU version, respecting the time constraint of $T_{cycle} = 20$ ms. In any case, notice that with the GPU implementation, occupancy grid map and Voronoi diagram transfer time, respectively 4.01 ms and 3.80 ms, have to be added to times shown in Table 1. Voronoi diagrams contain a more long-term information with respect to occupancy grid maps that contain more dynamical information. We synchronize the sending process so a new diagram with a new path is sent every 400 ms. Occupancy grid maps are sent every 200 ms. When the path planner cannot compute a trajectory in less than 20 ms, the vehicle controller uses an old command. If the GPU has to upload all maps it needs about $4 + 3$ ms. Then the remaining time is $20 - 7 = 13$ ms. As a consequence, the last configuration for which the GPU is able to compute a trajectory in *every* cycle is the one with $D = 6$ and $H = 4$. Nevertheless, most time-consuming configurations, such as the ones with $D = 5$, $H = 5$ (needed 15 ms) and $D = 4$, $H = 6$ (needed 18 ms), can still run within $T_{cycle} = 20$ ms in all cycles but the ones in which maps have to be updated.

**Table 1.** Comparing CPU and GPU wall-clock or elapsed times for several real-world scenarios. All times are reported in milli-seconds.

| D | H | # TRAJECTORIES | CPU | | | | GPU | | | |
|---|---|---|---|---|---|---|---|---|---|---|
| | | | M | $T_1$ | $T_2$ | Tot | M | $T_1$ | $T_2$ | Tot |
| | | | (KB) | (ms) | (ms) | (ms) | (KB) | (ms) | (ms) | (ms) |
| 6 | 3 | 216 | 10 | 15 | 4 | 19 | 16 | 2 | 2 | 4 |
| 5 | 4 | 625 | 32 | 26 | 7 | 33 | 51 | 3 | 2 | 5 |
| 6 | 4 | 1296 | 62 | 45 | 10 | 55 | 99 | 9 | 3 | 12 |
| 5 | 5 | 3125 | 156 | 77 | 14 | 86 | 250 | 11 | 4 | 15 |
| 4 | 6 | 4096 | 218 | 83 | 16 | 99 | 349 | 12 | 6 | 18 |
| 5 | 6 | 15,625 | 781 | 218 | 44 | 262 | 1250 | 38 | 18 | 46 |

Figure 14 finally shows that the quality of the results comparing CPU and GPU trajectories using the parameters introduced in Section 6.1. Figure 14 includes two sets of experiments. In all cases, as described in Section 6.3 we selected $T_{lookahead} = 3$ s and $T_{sim} = 20$ ms, whereas $D$ and $H$ are the one used in Table 1. Figure 14a–c represent experiments where the elk test is repeated with two different speeds, i.e., 25 m/s and 36 m/s. Figure 14d–f report experiments where the elk test is repeated with different distances between the two obstacles. As described in Section 6.3 the original distance is 75 m; now we use the 80%, 60%, 40%, and 20% of that value and a 25 m/s speed. The different histogram bars represent different values of $D$ and $H$ as reported in the picture caption. Overall, trajectory accuracy is maintained when the CPU and GPU are using the same setting. At the same time, as the GPU is faster, it is possible to use better parameter values with it. In those cases the starting distance, the root mean square error, and the minimum obstacle distance can be trimmed a bit more by creating trees with a higher number of levels or a higher degree.

As far as Figure 14b is concerned, notice that by increasing $D$ and $H$, we obtain smaller *RMSE* values with 25 m/s but larger with 36 m/s. This is due to the fact that with higher speeds, the expansion tree spans the space for longer distances and avoidance maneuvers start earlier. This is also confirmed by Figure 14a,c. In the first one, the safety distance is higher with 36 m/s than with 25 m/s. In the

second one, the minimum obstacle distance decreases with higher speeds. This behavior better reproduces a more realistic human driving style.

In Figure 14c the bar for $D = 5$ and $H = 5$ is higher than the one with $D = 4$ and $H = 6$. This is motivated by the consideration that the first splitting policy spans better the surrounding area, whereas the second one encompasses the space at farther distances.

As a final remark, notice that the closer the two obstacles get, the more the vehicle speed tends to decrease. We do not report evidence on this issue, but it has a strong impact on the minimum obstacle distance and some impact on the root mean square error.

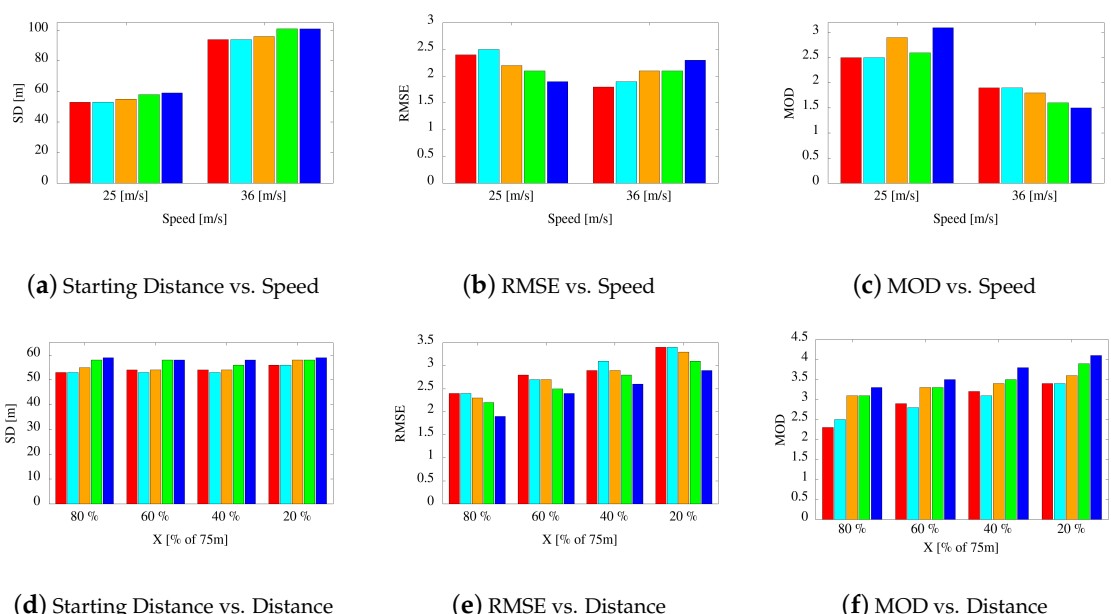

(**a**) Starting Distance vs. Speed    (**b**) RMSE vs. Speed    (**c**) MOD vs. Speed

(**d**) Starting Distance vs. Distance    (**e**) RMSE vs. Distance    (**f**) MOD vs. Distance

**Figure 14.** The elk test repeated with different speeds (**a**–**c**) and different distances between the two obstacles (**d**–**e**). The plots report the starting distance (SD) (**a**,**d**), the root mean square error (*RMSE*) (**b**,**e**), and the minimum obstacle distance (MOD) (**c**,**f**). The histograms report a comparison between CPU and GPU results. The red (first) column represent the CPU response with $D = 6$ and $H = 4$. All other colors represent the GPU response with: $D = 6$ $H = 4$ (second column, cyan), $D = 5$ $H = 5$ (third column, orange), $D = 4$ $H = 6$ (fourth column, green), and $D = 5$ $H = 6$ (fifth column, blue).

## 7. Conclusions and Future Works

Autonomous cars are supposed to become a reality in the next decade. A competitive autonomous car must acquire and fuse all data coming from environmental sensors and find a comfortable, minimum time collision free path in real time. Computation power and efficiency are then important issues to have real-time applications with reduced costs.

Local planning strategies represent the most sensible part of the entire path planning process, as local trajectories are subject to stringent mathematical constraints that prevent purely mathematical solutions. *Sample-based* planning techniques samples the configuration space into a set of finite motion goals, and do not need sophisticated mathematical approaches. *Randomized sample-based* planning algorithms are mostly based on *Rapidly-exploring Random Trees*, in which the driving idea is to iteratively expanded a random tree to drive the vehicle toward points selected randomly in close proximity of the trajectory.

This paper focuses on re-engineering a state-of-the-art randomized sampling-based motion planning method for an embedded many-core concurrent computation environment. The paper presents how to re-implement the original sequential algorithm using several concurrent CUDA

kernels. It shows how to enforce regularity, and how to appropriately store all data that have to be transferred from the CPU to the GPU (and vice-versa) and exchanged among different kernels.

We compare the original sequential algorithm with the highly parallel one, in terms of a few evaluation metrics (starting distance, root mean square error, and minimum obstacle distance), and in terms of wall-clock times. We prove that the accuracy of the original algorithm is essentially maintained when the algorithm is run with the same setting. Moreover, we prove that the GPU is able to obtain a 5x speed-up leaving the CPU free to work on any other task the designer may deem necessary on board. This speed-up can be used to obtain a fine-grained and denser space analysis, and a higher reactivity of the system in safety critical conditions.

One of the limits of the current approach is that many parameters of the original algorithm can be selected only in a static way. This strategy, albeit conservative, may be unsuited for rapidly changing driving conditions. As a consequence, as far as future works are concerned, one of the directions to improve the overall algorithm is to adopt the GPU to dynamically select and change the main parameters of the algorithm, such as the tree height or the splitting policy. Moreover, those parameters need to be better relate to the environment context, and the driving preferences, to reach safer and better planning trajectories.

As a final remark, for sure the planner needs more on-the-field experiments to check it on every-day conditions and its real applicability on the somehow very limited, and restricted, on-board hardware configurations.

**Author Contributions:** A.G. and F.S. developed the tool. M.G. defined the real situations on which to test it. S.Q. conceived and designed the experiments and analyzed the data. S.Q., P.C. and G.C. wrote the paper.

**Funding:** This research was funded by Magneti Marelli.

**Conflicts of Interest:** The authors declare any conflict of interest.

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
