# Peer review of "A Smart Many-Core Implementation of a Motion Planning Framework along a Reference Path for Autonomous Cars"

_electronics, doi:10.3390/electronics8020177_

Round 1
Reviewer 1 Report
This manuscript presents and discusses an implementation of a motion planner in a General Purpose Graphics Processing Units. In the point of view of robotics, none novel algorithm is presented to the community. This paper focuses in the use of parallel computing of well-known algorithms.
The paper is well organized and stresses an actual and emerging topic. Meanwhile, there are some points that should be improved:
The background and related works should address the A* path planning methodology.
Line 314: please describe in a deeper way the “RGBA image to encode 4 maps”.
How the obstacles are introduced into the system? (vision? Laser? sonar?...)
Even the experimental results could not be made publicly available, an interesting point that authors address in the paper is the consumption. The CPU and GPU wall-clock or elapsed times for several real-world scenarios are presented but the power consumption should be addresses and compared for both systems (hardware devices: CPU core i7 and GPGPU GEFORCE).
Typos:
Figure 9 caption: place -> placed
Figure 14: Please insert axis legends (x and y)
Author Response
We thank the Editor and the Reviewers for the careful reading and their good suggestions.
We addressed all comments in this new version of the paper.
Overall, we improved the overall style and quality.
All *main* modifications are highlighted in the paper by a *blue* color.
Details follow as direct answers to the Reviewers' comments.
> This manuscript presents and discusses an implementation of a motion
> planner in a General Purpose Graphics Processing Units.
> In the point of view of robotics, none novel algorithm is presented to
> the community.
> This paper focuses in the use of parallel computing of well-known
> algorithms.
>
> The paper is well organized and stresses an actual and emerging
> topic.
> Meanwhile, there are some points that should be improved:
>
> The background and related works should address the A* path planning
> methodology.
As described in Section 2.1 path planning is generally composed by a global planner, a decision maker, and a local planner.
We concentrate on the local planner, whereas A* is a global planner.
Anyhow, the new version of the paper includes the original reference and a brief description of the A* algorithm (see blue text on Page 1 and 2), as requested.
> Line 314: please describe in a deeper way the "RGBA image to encode
> 4 maps".
The idea is simple: As a texture pixel is represented on 4 floating-point values and each map pixel requires just 1 float, we use a texture to store 4 maps.
Anyhow, a complete description of our encoding strategy has been introduced in Section 5.2, page 10 and 11 (please, see blue text).
> How the obstacles are introduced into the system? (vision? Laser?
> sonar?...)
Magneti Marelli cars are usually equipped with vision cameras, GPS, radar and lidar sensors (sometimes even ultrasound sensors).
These data are given as inputs to a data fusion system generating the occupancy grids (and the local paths) used by our path planner.
We consider this topic outside the scope of the path planner and our paper as well.
Anyhow, we inserted a more accurate description of this topic in Section 4.1, right at the end of the terminology section (please, see blue text from line 268 on).
> Even the experimental results could not be made publicly available, an
> interesting point that authors address in the paper is the
> consumption. The CPU and GPU wall-clock or elapsed times for several
> real-world scenarios are presented but the power consumption should be
> addresses and compared for both systems (hardware devices: CPU core i7
> and GPGPU GEFORCE).
We agree with the reviewer that the issue of power consumption is an important one.
We added a brief explanation in the introduction (from line 91 on).
It includes a citation of a survey on power (and energy) estimation (and measurement) methods.
Roughly speaking, the adopted GPGPU (the NVIDIA GEFORCE GTX 970) has a maximum power consumption TDF (Thermal Design Power) about 3 times larger than the one of the adopted CPU (the Intel Core i7-6700 HQ).
Anyhow, real power consumption depends on many factors.
It is still debated in the scientific community as to whether GPUs are more energy and power consuming than CPU.
The verdict depends on the selected hardware platforms and on software implementation issues.
We are not considering the final hardware platforms in our work, as these are unknown (or undisclosed by Magneti Marelli so far).
We thus work on an industrial prototype including off-the-shelf and common CPUs and GPUs.
Although required for the final hardware/software architecture evaluation, CPU versus GPU comparison on power efficiency may thus be considered beyond the scope of this work.
> Typos:
> Figure 9 caption: place -> placed
> Figure 14: Please insert axis legends (x and y)
These problems have been rectified.
Reviewer 2 Report
This paper focuses on re-engineering an existing state-of-the-art sequential algorithm to obtain a CUDA-based GPGPU (General Purpose Graphics Processing Units) implementation. The paper compares the original sequential algorithm with the highly parallel one. The paper shows that the GPU is able to obtain a 5x speed-up leaving the CPU free to work on any other task the designer may deem necessary on board.
The reviewer has following comments. At line 319, there is the following sentence: The drawing sample procedure, in both CPU and GPU versions, uses the Voronoi diagram for the generation of reference points. Does this imply that reference points are generated along a Voronoi edge?
Algorithm 2 uses Voronoi diagram. However, Algorithm 1 does not use Voronoi diagram. Why does Algorithm 2 use Voronoi diagram while Algorithm 1 does not use it? Also, the paper is lack of definition for Voronoi diagram.
In general, Figures are lack of captions explaining the associated figures.
For instance, the caption of Fig.2 is lack of explanation. In Fig.2 , what is the meaning of black line which is parallel to the x- axis? In Fig.2, x-axis does not indicate time. Why did you plot time in Fig. 2? Fig.2 is the occupancy grid map seen from the above of the car.
Voronoi diagram is presented in Fig 6(b). However, Fig 6(b) is totally unclear to me. Where are obstacles? Voronoi edge is the set of points equidistant from two obstacles. In Alg. 2, "surf" seems to imply "surface memory". Is this correct?
The paper is hard to follow. At line 381, Figure 6 is explained. However, Figure 6 appears after line 424. Can you move Figure 6 close to line 381? Can you detail the caption of Figure 6 so that a reader can understand Figure 6 by just looking at the figure?
At line 412, "the second iteration of the loop at line 6 of Figure 2" does not make sense.
In Algorithm 1, N is missing at lines 4 and 12.
Author Response
We thank the Editor and the Reviewers for the careful reading and their good suggestions.
We addressed all comments in this new version of the paper.
Overall, we improved the overall style and quality.
All *main* modifications are highlighted in the paper by a *blue* color.
Details follow as direct answers to the Reviewers' comments.
> This paper focuses on re-engineering an existing state-of-the-art
> sequential algorithm to obtain a CUDA-based GPGPU (General Purpose
> Graphics Processing Units) implementation.
> The paper compares the original sequential algorithm with the highly
> parallel one.
> The paper shows that the GPU is able to obtain a 5x speed-up leaving
> the CPU free to work on any other task the designer may deem necessary
> on board.
>
> The reviewer has following comments.
>
> At line 319, there is the following sentence:
> The drawing sample procedure, in both CPU and GPU versions, uses the
> Voronoi diagram for the generation of reference points.
> Does this imply that reference points are generated along a Voronoi
> edge?
Our local planner receives from the data fusion module, the Voronoi diagram, with the reference path, and the occupancy grid.
The target of our local planner is to follow the path respecting the occupancy grid map.
The Voronoi Diagram is used to efficiently find goal points for our tree expansion.
It is not used to represent or to manage obstacles.
These topics are described at the beginning of Section 5.2,
This part has been heavily rewritten (please see the blue sections from line 327 to line 371) to make it more complete and more understandable.
> Algorithm 2 uses Voronoi diagram.
> However, Algorithm 1 does not use Voronoi diagram.
> Why does Algorithm 2 use Voronoi diagram while Algorithm 1 does not
> use it?
> Also, the paper is lack of definition for Voronoi diagram.
Voronoi diagrams are used by both algorithms (Algorithm 1, the reference one running on CPUs, and Algorithm 2, the concurrent version running on GPUs), and by the original implementation
by Schwesinger et al. (old reference [9], now reference [10]) as well.
The reason for this is quite simple.
When the car is not on the path and it is physically impossibleb to select a goal on the path, the algorithm selects a feasible goal and then it approximates such a goal with the closest point on the
path.
Finding the closest point on the path to a given point in all steps would be really time consuming.
This is where Voronoi diagrams get handy, as they save a lot of computation time creating "Voronoi cells" of points closer to the desired "Voronoi seeds".
A proper mathematical description of the Voronoi diagram has been introduced in Page 11 (see please caption 3).
> In general, Figures are lack of captions explaining the associated
> figures.
All captions have been revised to be more detailed and precise.
> For instance, the caption of Fig.2 is lack of explanation.
> In Fig.2 , what is the meaning of black line which is parallel to the
> x-axis?
> In Fig.2, x-axis does not indicate time. Why did you plot time in Fig.2?
> Fig.2 is the occupancy grid map seen from the above of the car.
The Reviewer is right and we were too vague.
Figure 2 is the occupancy grid map seen from the above of the car thus the x-axis represents space.
Anyway, as the car is moving along the x axis space positions correspond to times,
We have redrawn the picture to indicate two x-axis (for the space and the time representations) and we have rewritten its caption.
> Voronoi diagram is presented in Fig 6(b).
> However, Fig 6(b) is totally unclear to me. Where are obstacles?
> Voronoi edge is the set of points equidistant from two obstacles.
We use the Voronoi maps to compute the points on the desired path closer to the goal points.
We do not use them to represent obstacles.
Obstacles are represented on the occupancy grids.
Occupancy grids and Voronoi maps are now better described one after the other starting from line 338 on Page 10 and 354 on Page 11.
Please see previous responses as well.
> In Alg. 2, "surf" seems to imply "surface memory".
> Is this correct?
Yes, it is correct.
Surface memories are introduced starting from line 334 (old line 309), and recalled describing Algorithm 2 (from line 386, old line 346).
> The paper is hard to follow.
> At line 381, Figure 6 is explained.
> However, Figure 6 appears after line 424.
> Can you move Figure 6 close to line 381?
> Can you detail the caption of Figure 6 so that a reader can understand
> Figure 6 by just looking at the figure?
The caption has been changed (actually almost all captions have been modified, please see the blue text in all captions).
All figures have been moved as close as possible to their textual descriptions.
> At line 412, "the second iteration of the loop at line 6 of Figure 2"
> does not make sense.
We meant that the second part of Figure 5 (the one with h=1) represents the situation of our data structures when the "for" cycle at line 6 of Algorithm 2 iterates for the second time (with h=1, in fact),
Anyhow, we have changed the text to describe this concept better (please, see the blue paragraph starting at line 451).
> In Algorithm 1, N is missing at lines 4 and 12.
These problems have been rectified.
Round 2
Reviewer 2 Report
I am satisfied with this paper. Thank you.